# Optimal Pattern Detection Tree for Symbolic Rule-Based Classification

**Young-Chae Hong**                                                    *hongych@amazon.com*
*Amazon*
*Seattle, USA*

**Yangho Chen**                                                        *yanghoc@amazon.com*
*Amazon*
*Seattle, USA*

**Reviewed on OpenReview:** *https://openreview.net/forum?id=RJ6eMDcDCv*

## Abstract

Pattern discovery in data plays a crucial role across diverse domains, including healthcare, risk assessment, and machinery maintenance. In contrast to black-box deep learning models, symbolic rule discovery emerges as a key data mining task, generating human-interpretable rules that offer both transparency and intuitive explainability. This paper introduces the Optimal Pattern Detection Tree (OPDT), a rule-based machine learning model based on novel mixed-integer programming to discover a single optimal pattern in data through binary classification. To incorporate prior knowledge and compliance requirements, we further introduce the Branching Structure Constraints (BSC) framework, which enables decision makers to encode domain knowledge and constraints directly into the model. This optimization-based approach discovers a hidden underlying pattern in datasets, when it exists, by identifying an optimal rule that maximizes coverage while minimizing the false positive rate due to misclassification. Our computational experiments show that OPDT discovers a pattern with optimality guarantees on moderately sized datasets within reasonable runtime.

## 1 Introduction

Data mining is the process of discovering and extracting hidden patterns from datasets to gain valuable insights and support decision-making. In particular, symbolic rule discovery is an important data mining task that generates human-interpretable rules in a natural and intuitive manner. However, recent developments in artificial intelligence (AI) and machine learning (ML) have driven attention toward black-box models, particularly deep learning approaches. This trend has led, in turn, to a growing demand for developing algorithms that inherently learn interpretable white-box models, especially in high-stakes domains such as healthcare, personalized medicine, criminal justice, and financial risk assessment, where decisions can significantly impact human lives (Rudin, 2019; Rudin et al., 2022). The exponential growth in healthcare data collection, driven by advanced monitoring technologies and digital health systems, has resulted in a growing volume of biological and medical data at unprecedented speed and scale (Luo et al., 2016). While the tremendous amounts of data—collected from electronic health records or monitoring devices can be utilized for more effective and enhanced clinical decision making (Rea et al., 2012), it poses significant challenges in efficiently analyzing and extracting useful information and actionable insights from such overwhelming volumes of data (Najarian et al., 2013).

Over the last decade, various pattern recognition techniques have been developed for medical decision support systems to help clinicians effectively utilize the overwhelming amount of healthcare data. These techniques are applied to biomedical data for automated clinical diagnosis or therapeutic support, integrating data-driven knowledge and patient-specific information to enhance cost-effective healthcare delivery (Moja et al.,

2014). The development of novel pattern recognition methods and algorithms with high performance, in terms of accuracy and time complexity, improves healthcare delivery by allowing clinicians to make a better-informed and timely decision (Asgari et al., 2019). To take reliable actions in high-stakes domains, the patterns extracted from data must be understood by human domain experts, unlike the opaque solutions provided by black-box ML approaches (Rudin, 2019). If domain experts do not fully comprehend the reasoning behind these patterns, there is a risk of making misinformed or potentially harmful decisions in life-threatening situations. Therefore, developing human-readable rules is crucial for addressing emerging challenges in critical decision-making processes and ensuring accountability, transparency, and trust in automated systems.

Rule-based machine learning (RBML) models offer a promising solution by combining the interpretability of traditional rule-based systems with the automated learning capabilities of ML approaches. Unlike conventional black-box ML models, RBML generates explicit decision rules that can be readily understood and validated by domain experts. Furthermore, for high-risk tasks requiring accountability, not only is interpretability crucial, but models must also comply with domain expert-specified constraints to be trustworthy (Nanfack et al., 2022). For example, learned patterns may not make sense from a medical or clinical perspective, as algorithms typically consider only information extracted from medical datasets without incorporating domain knowledge (López-Vallverdú et al., 2007). Therefore, ML models can become more trustworthy and reliable if they incorporate additional domain knowledge in the form of constraints. However, existing RBML methods are designed to extract a ruleset (i.e., a collection of multiple rules) rather than a single optimal pattern in data. In many real-world scenarios, such as an epidemic outbreak, credit fraud, or intrusion detection, positive samples originate from a single underlying cause, and a single high-precision rule is both sufficient and more actionable than a complex ruleset. To the best of our knowledge, no prior method has conducted research on extracting a single optimal pattern.

This paper proposes a novel algorithm to extract a single rule through binary classification that adheres to user-defined structural constraints, ensuring high interpretability while incorporating prior domain knowledge. We develop the Optimal Pattern Detection Tree (OPDT) based on optimal decision trees with *branching structure constraints* that control: 1) the tree's topology through split decisions, and 2) the feature group assignments for each branch node. This optimization-based rule extracting approach discovers a hidden underlying pattern in datasets, when it exists, by identifying an optimal rule that maximizes coverage while minimizing the false positive rate due to misclassification. The remainder of the paper is structured as follows. In Section 2, we discuss the related work regarding the interpretable RBML and optimal decision tree (ODT). In Section 3, we present our proposed approach, OPDT to extract an optimal pattern from a dataset. In Section 4, we demonstrate the performance of the OPDT over different real-world datasets from the UCI repository and compare it with other RBML methods.

## 2 Related Literature

### 2.1 Rule-Based Machine Learning

Among advanced AI and ML technologies, the ability to extract valuable insights from complex datasets has become crucial for various fields, including healthcare, criminal justice, and financial risk assessment. However, the lack of interpretability in ML models can potentially lead to adverse or even life-threatening consequences (Ahmad et al., 2018). There is a growing demand for interpretable ML that allows end-users to understand the logic and reasoning behind predictions, especially in domains such as healthcare and criminal justice systems. This demand for improved interpretability has also increased to mitigate the risk of making unjustifiable decisions. In particular, studies in domains like healthcare and criminal justice have revealed that ML systems can systematically exhibit unfair biases (Burrell, 2016). Thus, interpretability is needed to ensure such systems are free from bias (Hajian et al., 2016). Recent research has focused on developing explainable or interpretable AI models (Murdoch et al., 2019; Vilone & Longo, 2021) because model interpretability and explainability are crucial for clinical and healthcare practices (Ahmad et al., 2018; Kolyshkina & Simoff, 2021), where misclassification costs can be potentially high. To address this, data mining techniques have emerged to discover interpretable rules by generating symbolic rules that humans can readily understand. By providing understandable patterns or rules from training data, we can involve

human experts such as doctors or nurses, allowing them to review AI model results and provide feedback or corrections. Consequently, this approach can improve the reliability of life-critical decision-making in healthcare.

Unlike deep learning-based models that often operate as black boxes, RBML is an algorithm that induces a set of symbolic rules from training data. RBML automatically identifies regularities or patterns that can be expressed in the form of "IF-THEN" statements. Here, the left hand side ("IF" part) is called the rule antecedent or condition, which can be expressed in logical operators like disjunction, conjunction, and negation. The right hand side ("THEN" part) of a rule is called the rule consequent or conclusion. In particular, rule conditions represented in disjunctive normal form (DNF: OR-of-ANDs) or conjunctive normal form (CNF: AND-of-ORs) are intuitively understandable to humans as they mirror natural reasoning processes. For example, a medical diagnosis rule might state: "IF (patient age > 65 AND white blood cell count > 11,000) OR (body temperature > 100 AND patient has chest pain) THEN (further diagnostic tests for potential infection)". A natural choice for interpretability is to represent ML output as logical operators like disjunction, conjunction, and negation with the IF-THEN structure, and these can be expressed through rule sets, decision trees, and decision lists. RBML's full transparency and intuitive explainability make it particularly valuable in applications where decision-making processes must be clear and explainable. The principles of RBML are most commonly implemented through the form of rule-based classifiers in modern ML applications. Rule-based classifiers can be seen as a generalization of decision tree classifier since the obtained rules do not need to be represented in the form of a tree, thus being more flexible (Palliser-Sans, 2021). Unlike trees, DNF rules need not be mutually exclusive. These rules are potentially more compact and predictive than trees. Regarding the induced rules, it uses conjunctive antecedents: "IF condition_01 AND condition_02 AND ... THEN conclusion". Each individual condition, also called a selector, is formed by an attribute-value pair, where the pair refers to a specific feature of the data and its corresponding value. Rule-based classifiers have already demonstrated successful application across diverse domains, including medical diagnosis (Asgari et al., 2019), financial fraud detection (Ali et al., 2022), intrusion detection (Lee et al., 1999), and machine failure diagnosis in manufacturing systems (Jiang'hong & Xiao'li, 2009).

Research on RBML has introduced various approaches to extract rules from data. First, Rules Extraction System (RULES) (Pham & Aksoy, 1995) is a simple algorithm for extracting a set of classification rules from a set of training instances given a set of classes. It follows a general-to-specific approach and enforces perfect precision (100%) in the generated rules unless there are inconsistencies in the data. RRULES (Palliser-Sans, 2021) was proposed as an optimization of RULES by focusing on two key points: the mechanism to detect irrelevant rules and the stopping condition. PRISM (Cendrowska, 1988) introduced a unique approach of generating rules by selecting examples of a specific class and iteratively adding conditions until the obtained rule has perfect precision. As a non-ordered and non-incremental algorithm, PRISM builds rules from general to specific patterns, with rule order being irrelevant for predictions. Interpretable Decision Sets (IDS) (Lakkaraju et al., 2016) presented a method for generating decision sets which are sets of independent if-then rules. Since each rule can be applied independently, decision sets are simple, concise, and easily interpretable compared to a decision list. Bayesian Rule Sets (BRS) (Wang et al., 2017) introduced a novel Bayesian framework for learning rule sets that uses a generative model to incorporate prior knowledge about interpretable rules while maintaining strong predictive performance. Incremental Reduced Error Pruning (IREP) (Fürnkranz & Widmer, 1994) addressed the computational inefficiencies of standard reduced error pruning in rule learning by integrating the pruning process directly into rule learning rather than as a post-processing step. This innovation laid the foundation for RIPPER (Cohen et al., 1995), which demonstrated that incremental pruning could achieve comparable or better results than standard pruning while being substantially more efficient. RuleFit (Friedman & Popescu, 2008) extracts rules from an ensemble of trees by automatically detecting interaction effects in the form of decision rules and builds a weighted combination of these rules using L1-regularized optimization over the weights (Friedman & Popescu, 2004). SkopeRules (Goix et al., 2020) is based on RuleFit's approach, utilizing a Random Forest model to fit class labels. The two methods differ only in their rule pruning strategy: RuleFit uses a linear model whereas SkopeRules heuristically deduplicates overlapping rules.

## 2.2 Optimal Classification

In recent years, researchers have increasingly applied discrete optimization techniques to learn optimal decision trees in ML problems. The problem of learning an optimal decision tree is NP-hard (Laurent & Rivest, 1976), which has led to the widespread adoption of greedy heuristics such as Classification and Regression Trees (CART) (Breiman et al., 1984), which constructs univariate classification trees. However, over the last decades, there has been an overall 800 billion-fold speedup in the computational power of optimization solvers (Bertsimas & Dunn, 2017). This astonishing increase in optimization solver performance has made it possible to apply modern mixed-integer optimization (MIO) methods to solve optimal decision trees. An overview of recent developments in optimization techniques for ML is presented in Bottou et al. (Bottou et al., 2018) and Gambella et al. (Gambella et al., 2021).

Recently, Bertsimas & Dunn (2017) introduced optimal classification tree (OCT), which is a mixed-integer programming (MIP) formulation to learn optimal decision trees given a fixed depth. The constraints and variables of the formulation can be decomposed into three sets: those defining the structure of the tree through split decisions at each node, those routing data samples from root to leaf nodes, and those counting misclassifications. The objective function balances minimizing training set misclassifications against tree complexity to preserve interpretability (Ales et al., 2024). The MIP model supports both axis-aligned and oblique splits with exponential complexity in tree depth, and can handle both continuous and categorical features. Günlük et al. (2021) proposed an alternative formulation for optimal decision trees that specializes in categorical features. Their formulation exploits the combinatorial structure of categorical variables, enabling branching on subsets of categorical feature values. However, both models can easily become intractable as the size of training dataset grows. To address this issue, Verwer & Zhang (2019) proposed BinOCT, which represents decision thresholds through binary encoding, drastically reducing the number of decision variables. The number of decision variables is largely independent of the training set size. It only depends logarithmically on the number of unique feature values. To further improve scalability, Firat et al. (2020) introduced a new formulation based on root-to-leaf paths for fixed-depth trees, which is solved using a column generation-based heuristic. Finally, Aghaei et al. (2024) introduced a flow-based MIP formulation that represents optimal classification trees as a maximum flow problem. The flowOCT formulation avoids big-M constraints, resulting in stronger linear programming (LP) relaxations compared to previous OCT models. However, it only works with binary features.

In addition to classification decision trees, recent work has explored discrete optimization techniques to learn boolean rules and rule sets. Su et al. (2016) developed an integer programming (IP) formulation for interpretable two-level boolean rules, expressed in either CNF or DNF, with a predefined maximum number of clauses, enabling controlled model complexity and enhanced transparency in classification tasks for high-stakes domains such as healthcare and law. Lawless et al. (2023) proposed an IP framework using column generation to learn sparse and interpretable boolean rule sets in DNF form, while incorporating fairness constraints.

# 3 Methods

This section presents a structure-constrained decision tree approach to identify patterns in datasets while adhering to prior domain knowledge and structural constraints defined by domain experts. In rule-based classification, any pattern or rule can be expressed through a nested if-then statement since their logical structure inherently consists of conditions (IF clause) and consequences (THEN clause). On the other hand, decision trees effectively extract meaningful patterns from data and transform them into interpretable rules by linearizing conditions along paths from root to leaf nodes. Therefore, we utilize a decision tree approach that discovers classification patterns through nested if-then-else structures, ensuring all possible combinations of conditions, including negation scenarios.

Most business domains have domain-specific prior knowledge about their data. For example, in medical diagnosis, physicians know that elevated blood pressure and high cholesterol levels are key risk factors for cardiovascular disease. Similarly, in diabetes prediction, medical experts rely on crucial diagnostic indicators such as body mass index (BMI), fasting blood glucose levels, and age. Furthermore, domain

experts recognize that features can be naturally grouped based on their relevance. In cardiovascular disease assessment, for instance, blood test results (cholesterol levels, triglycerides, and blood glucose) form one group, while vital signs (blood pressure, heart rate, and body temperature) form another. Even when domain knowledge is unavailable, various ML methods can extract prior information about features. For example, SHAP (SHapley Additive exPlanations) is a model-agnostic method that explains the contribution of each feature to model outputs (Lundberg & Lee, 2017). Similarly, Boruta is a feature selection algorithm that identifies which features are statistically significant or relevant to the outcome variable (Kursa & Rudnicki, 2010). Additionally, feature importance in decision trees quantifies the relative contribution of each input feature to the model's predictions. Therefore, we can reasonably assume two types of prior knowledge: 1) feature groupings based on feature correlation and 2) feature impact on the target variable based on feature importance. Given the prior knowledge, we propose a structure-constrained decision tree to identify optimal patterns in data. This paper develops an optimization-based model that utilizes prior information from either domain experts or ML algorithms.

### 3.1 Structure Constrained Decision Tree

A structure-constrained decision tree (SCDT) refers to a decision tree where the learning process is guided and restricted by specific constraints imposed on its structure. These constraints aim to improve the interpretability, fairness, compliance, or other desired properties of the resulting tree. In pattern discovery, we incorporate prior domain knowledge into the decision tree through structural constraints, following three steps to detect patterns in data. First, given the prior knowledge from domain experts or feature relevance determined by ML algorithms, we define a feature group as a subset of features that captures a distinct semantic or conceptual aspect of the classification problem, where feature groups are not required to be disjoint and may overlap. For instance, in diabetes diagnosis, laboratory measurements (fasting blood glucose, HbA1c levels) and patient characteristics (age, BMI, family history) can be formed into separate groups, reflecting their distinct roles in diagnosis. As shown in Figure 1, our framework allows feature groups to be overlapping, providing flexibility in capturing features that may belong to multiple semantic or conceptual categories. Additionally, we define $G_A$ as a special feature group containing all features, which serves as a default group when prior domain knowledge is insufficient to establish meaningful feature groupings. Second, we define the topology of the binary tree using the feature groups, which restricts feature availability at each branching node and the depth of the tree. For example, in medical diagnosis, blood test results might be restricted to upper-level nodes while patient symptoms are considered at lower levels, following clinical diagnostic procedures. Finally, we run the SCDT where feature selection at each branching node is guided by predefined feature groups. This approach ensures the learning process follows the branching constraints imposed on the decision tree.

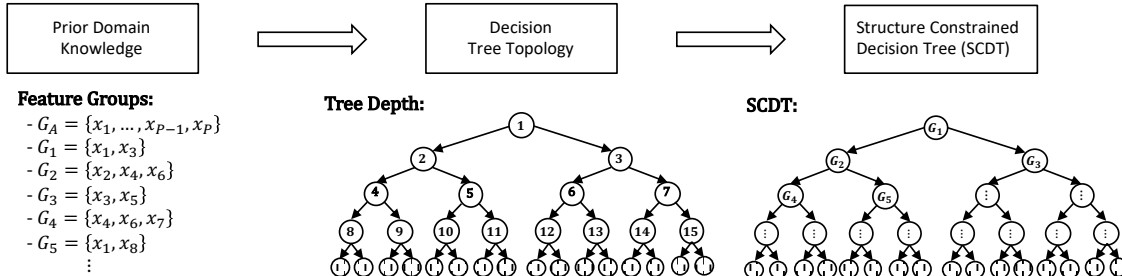

Figure 1: Structure Constrained Decision Tree (SCDT). The tree uses a breadth-first indexing scheme where nodes are numbered sequentially from the root (index 1) through internal nodes to leaf nodes. Feature groups $(G_A, G_1, G_2, \dots)$ can be overlapping, where $G_A$ denotes a special feature group containing all features, used when prior domain knowledge for defining feature groups is unavailable.

### 3.2 Optimal Pattern Detection Tree

In this section, we first introduce the volume-impurity index that measures coverage volume and misclassification by a rule and then present the basic concepts of OPDT. Subsequently, we present the IP

formulation of OPDT. Finally, we describe branching structure constraints on OPDT and their enhancements to improve computational efficiency.

**Metrics**  Prior RBML methods employ heuristic approaches with no optimality guarantees, focusing on precision maximization while treating coverage as a byproduct (Cendrowska, 1988), balancing precision with rule complexity such as the number of rules and conditions per rule (Lakkaraju et al., 2016), adopting a two-phase strategy that prioritizes coverage in the growing phase before shifting to precision improvement through pruning on separate validation sets (Fürnkranz & Widmer, 1994; Cohen et al., 1995), or balancing accuracy and interpretability through Bayesian likelihood and prior terms (Wang et al., 2017).

Unlike these traditional implicit or sequential evaluation approaches employed in heuristic rule learning, we define a novel metric specifically designed for optimization-based frameworks that explicitly balances coverage volume with misclassification control through the weight parameter $w$, allowing domain experts to specify misclassification tolerance directly within the objective function before model training. Specifically, we propose a volume-impurity (VI) index that balances classification volume and classification impurity, defined as:

$$\text{volume-impurity (VI) index} = \text{``volume gain''} - w \times \text{``misclassification loss''}$$

where $w$ is the weight controlling precision.

In rule discovery, volume gain refers to the size of samples covered by the rule, while misclassification loss represents the size of incorrectly classified samples among the covered samples. The goal of OPDT is to identify a rule (i.e., root-to-leaf path in a decision tree) that maximizes the VI index in the leaf node.

**Approach**  The primary focus in rule discovery is identifying patterns and regularities within a dataset. In this paper, we aim to identify an optimal rule that maximizes coverage volume while controlling misclassification through a given weight. We utilize a decision tree approach to discover this optimal rule, where each rule corresponds to a root-to-leaf path that maximizes the VI index through nested if-then-else structures handling negation scenarios while incorporating prior knowledge via structural constraints. As shown in Figure 2, given feature groups (e.g., $G_1, G_2, \ldots, G_D$) from prior domain knowledge or heuristic algorithms, we initially assume a fixed length of rule conditions, allowing us to assign feature groups to specific depths in the chain of rule conditions (e.g., $\{G_1 - G_2 - \cdots - G_D\}$) under the SCDT framework. While SCDT allows different feature groups at each branching node as shown in Figure 1, OPDT enforces the same feature group at each level of the decision tree to systematically explore all negation combinations (i.e., $\geq$ or $<$ for each feature within a feature group) across the chain of rule conditions. This determines the topology of the nested if-then-else tree. Using discrete optimization, we formulate our rule learning framework as a decision tree where leaf nodes are measured by the VI index and find rule conditions that maximize this value, yielding the optimal pattern among all possible combinations of rule conditions. We call this approach the Optimal Pattern Detection Tree.

**Mathematical Formulation**  We use discrete optimization to solve the OPDT problem. Since OPDT aims to discover rule conditions rather than making predictions based on majority class of training samples in leaf nodes, the approach discards most nodes except for the node with maximal VI value. It extracts splitting conditions along the path from root to the leaf node with maximal VI value to form the IF part of the decision rule. The learned rule is represented as a chain of conditions and a label prediction. When a sample meets all conditions in the IF part, we say the rule fires, and we define such a sample as covered by the rule. The rule is then evaluated by two metrics: coverage (measured by fired volume) and precision (measured by misclassification loss). We formulate OPDT as a mixed-integer program as follows:

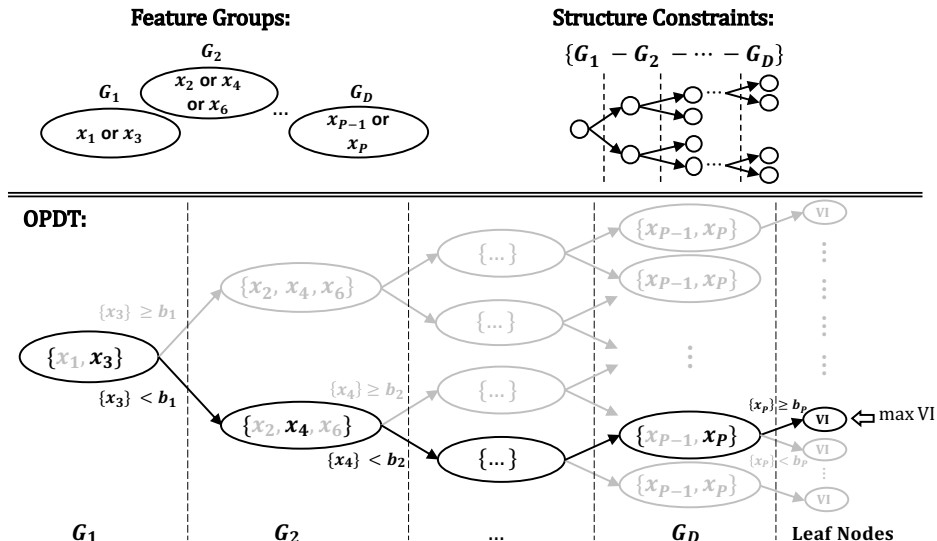

Figure 2: Optimal Pattern Detection Tree (OPDT). All nodes at the same depth are restricted to use identical feature groups for the chain of rule conditions $\{G_1 - G_2 - \cdots - G_D\}$.

Table 1: Sets for OPDT

| Symbol | Definition |
|---|---|
| $P$ | set of features in a dataset with one-hot encoded categorical variables |
| $N$ | set of samples in a dataset |
| $x_i$ | $i$th sample in a dataset, $i \in N$ |
| $y_i$ | $i$th sample's label in a dataset, $i \in N$ |
| $K$ | set of class label |
| $k$ | class label, $k \in K$ |
| $Y_{ik}$ | one-hot encoded matrix on $y$, $\mathbf{1}\{y_i = k\}$, $\forall k \in K, i \in N$ |
| $D$ | maximum depth of the optimal decision tree |
| $T$ | maximum possible nodes by tree with a depth of $D$, $T = 2^{(D+1)} - 1$ |
| $T_B$ | set of branch nodes with split $a^T x < b$, $T_B = \{1, \ldots, \lfloor T/2 \rfloor\}$ |
| $T_L$ | set of leaf nodes with class prediction, $T_L = \{\lfloor T/2 \rfloor + 1, \ldots, T\}$ |
| $T_{obj}$ | set of target leaf nodes where the VI index is evaluated in the objective function, $T_{obj} \subseteq T_L$ |
| $t$ | index of each node, $t \in \{1, \ldots, T\}$ |
| $w$ | weight on misclassified samples in a node, $w \geq 1$ |
| $p(t)$ | parent node of node $t$ |
| $A(t)$ | set of ancestors of node $t$, $A(t) = A_L(t) \cup A_R(t)$ |
| $A_L(t)$ | set of left-branch ancestors of node $t$ whose left branch has been followed on the path from the root node to node $t$ |
| $A_R(t)$ | set of right-branch ancestors of node $t$ whose right branch has been followed on the path from the root node to node $t$ |
| $E_L(t)$ | set of all leaf nodes from a left child of node $t$ |
| $E_R(t)$ | set containing two rightmost leaf nodes, one from each child (left and right) of node $t$ |
| $B(\text{split})[t]$ | binary value indicating whether branching node $t$ in $T_B$ contains a split, $|B(\text{split})| = |T_B|$ |
| $B(\text{group})[t]$ | feature group assigned to branching node $t$ in $T_B$, $|B(\text{group})| = |T_B|$ |
| $L(\text{target})[t]$ | binary value indicating whether leaf node $t$ in $T_L$ is a target leaf node, $|L(\text{target})| = |T_L|$ |

Table 2: Variables for OPDT

| Symbol | Definition |
|---|---|
| $a_{pt}$ | slope element of split applied at node $t$, $\forall p \in P, \forall t \in T_B$ |
| $\vec{a}_t$ | slope vector of split applied at node $t$, where $\vec{a}_t = (a_{1t}, a_{2t}, \ldots, a_{|P|t})$, $\forall t \in T_B$ |
| $b_t$ | intercept of split applied at node $t$, $\forall t \in T_B$ |
| $d_t$ | $\mathbf{1}\{$node $t$ applies a split$\}$, $\forall t \in T_B$ |
| $z_{it}$ | $\mathbf{1}\{x_i$ is in node $t\}$, $\forall i \in N, \forall t \in T_L$ |
| $l_t$ | $\mathbf{1}\{$leaf $t$ contains any sample$\}$, $\forall t \in T_L$ |
| $N_t$ | total number of samples in node $t$, $\forall t \in T_L$ |
| $N_{kt}$ | number of samples of label $k$ in node $t$, $\forall k \in K, \forall t \in T_L$ |
| $c_{kt}$ | $\mathbf{1}\{$the most common labels in node $t$ is $k\}$, $c_{kt} = \mathbf{1}\{\arg\max_{k \in K}\{N_{kt}\} = k\}$, $\forall k \in K, \forall t \in T_L$ |
| $L_t$ | the number of misclassified samples ($L_t = N_t - M_t$) in the leaf node $t$, where $M_t$ is total number of the most common labels in node $t$, $M_t = \max_{k \in K} N_{kt}$, $\forall t \in T_{obj}$ |
| $I_t$ | volume-impurity (VI) value calculated as the total number of samples minus the misclassified samples in a leaf node, $I_t = \{N_t - wL_t\}, \forall t \in T_{obj}$ |
| $I_{max}$ | maximum number of $I_t$ over all target leaf nodes, $I_{max} = \max_{t \in T_{obj}} I_t$ |
| $q_t$ | binary variable indicating whether leaf node $t$ has the maximum VI index, $\forall t \in T_{obj}$ |

$$\max I_{max} \tag{1}$$

subject to:

$$\sum_{p=1}^{P} a_{pt} = d_t, \quad \forall t \in T_B \tag{2}$$

$$d_t \leq d_{p(t)}, \quad \forall t \in T_B \setminus \{1\} \tag{3}$$

$$l_s \leq d_t, \quad \forall t \in T_B, \forall s \in E_L(t) \tag{4}$$

$$l_s \geq d_t, \quad \forall t \in T_B, \forall s \in E_R(t) \tag{5}$$

$$z_{it} \leq l_t, \quad \forall i \in N, \forall t \in T_L \tag{6}$$

$$N_t = \sum_{i=1}^{n} z_{it}, \quad \forall t \in T_L \tag{7}$$

$$\sum_{t \in T_L} z_{it} = 1, \quad \forall i \in N \tag{8}$$

$$\vec{a}_s^T(x_i + \vec{\epsilon}) \leq b_s + (1 + \epsilon_{max})(1 - z_{it}),$$
$$\forall i \in N, \forall t \in T_L, \forall s \in A_L(t) \tag{9}$$

$$\vec{a}_s^T x_i \geq b_s - (1 - z_{it}),$$
$$\forall i \in N, \forall t \in T_L, \forall s \in A_R(t) \tag{10}$$

$$\sum_{k=1}^{K} c_{kt} = l_t, \quad \forall t \in T_L \tag{11}$$

$$N_{kt} = \sum_{i=1}^{n} Y_{ik} z_{it}, \quad \forall k \in K, t \in T_L \tag{12}$$

$$L_t \geq N_t - N_{kt} - M(1 - c_{kt}),$$
$$\forall k \in K, \forall t \in T_{obj} \tag{13}$$

$$L_t \leq N_t - N_{kt} + M c_{kt},$$
$$\forall k \in K, \forall t \in T_{obj} \tag{14}$$

$$I_{max} \geq N_t - wL_t, \quad \forall t \in T_{obj} \tag{15}$$

$$I_{max} \leq N_t - wL_t + M(1 - q_t), \qquad \forall t \in T_{obj} \tag{16}$$

$$\sum_{t \in T_{obj}} q_t = 1 \tag{17}$$

$$d_t \in \{0, 1\}, \quad \forall t \in T_B \tag{18}$$

$$a_{pt} \in \{0, 1\}, \quad \forall p \in P, \forall t \in T_B \tag{19}$$

$$0 \leq b_t \leq d_t, \quad \forall t \in T_B \tag{20}$$

$$z_{it} \in \{0, 1\}, \quad \forall i \in N, \forall t \in T_L \tag{21}$$

$$l_t \in \{0, 1\}, \quad \forall t \in T_L \tag{22}$$

$$c_{kt} \in \{0, 1\}, \quad \forall k \in K, \forall t \in T_L \tag{23}$$

$$N_t \in \mathbb{N}, \quad \forall t \in T_L \tag{24}$$

$$N_{kt} \in \mathbb{N}, \quad \forall k \in K, \forall t \in T_L \tag{25}$$

$$L_t \in \mathbb{N}, \quad \forall t \in T_{obj} \tag{26}$$

$$q_t \in \{0, 1\}, \quad \forall t \in T_{obj} \tag{27}$$

$$I_{\max} \in \mathbb{R} \tag{28}$$

Our OPDT formulation adopts three fundamental constraints from Bertsimas & Dunn (2017)'s OCT: tree structure constraints (split decisions at branching nodes), sample routing constraints (directing samples from root to leaf), and misclassification constraints (counting misclassified samples at each leaf node). On top of these, we introduce constraints that force samples to rightmost leaf nodes when parent nodes do not split, correcting sample routing behavior for accurate VI evaluation at branching-terminated parent nodes, and constraints that linearize the logical condition for maximal VI selection among leaf nodes. The objective (1) maximizes volume on one of the leaf nodes while minimizing misclassification loss through the VI index, which combines volume and misclassification loss with weight $w$. Constraint (2) ensures that a slope variable $a_{pt}$ for a branching node can be activated if the node applies a split. Constraint (3) allows a branch node to split only if its parent node splits. Constraint (4) ensures that all leaf nodes from a left child of a branching node have zero samples if the node doesn't split, forcing samples to be assigned to the rightmost leaf nodes. Constraint (5) ensures that the two rightmost leaf nodes from both left and right children of a branching node must be active if the branching node splits. Constraint (6) guarantees that samples can be assigned to a leaf node if the node is active. Constraint (7) counts the total number of samples at leaves. Constraint (8) forces each sample to be assigned to exactly one leaf. Constraints (9 and 10) formulate the branching rules at each node, where $\vec{\epsilon}$ denotes a vector of sufficiently small positive values $\epsilon_p$ for each feature $p \in P$ and $\epsilon_{\max} = \max_p \epsilon_p$. These $\epsilon_p$ and $\epsilon_{\max}$ arise from a technical necessity in MIP formulations: strict inequality constraints of the form $a_t^\top x_i < b_t$ are not directly supported by MIP solvers and must be converted to non-strict form. Practically, $\epsilon_p$ can be any sufficiently small positive number that does not cause numerical instabilities in the MIP solver. On the other hand, the largest valid value of $\epsilon_p$ is the smallest non-zero distance between adjacent values of feature $p$ in the data. We use the largest valid value of $\epsilon_p$ for numerical stability. The parameter $\epsilon_{\max}$ acts as the big-M constant since the maximum possible value of $a_t^\top (x_i + \vec{\epsilon}) - b_t$ is $1 + \epsilon_{\max}$. Constraints (13 and 14) represent misclassification loss in each target leaf node, where $M$ denotes a big-M constant. Here, we take $M = |N|$ as a sufficiently large value since the misclassification loss at any leaf node is bounded by $|N|$. Constraints (15, 16, and 17) represent the maximum value of the VI index over all target leaf nodes, where the largest valid value of $M$ is $w|N|$. Finally, decision variables (18), (19), and (20) specify the branching structure, where $d_t$ indicates whether node $t$ has a split, $a_{pt}$ selects features for the split, and $b_t$ defines the threshold for split $a^T x < b$ at each branch node. Variables (21), (22), and (23) determine sample assignments to leaf nodes, leaf node activation, and class predictions, respectively. Count variables (24), (25), and (26) track the number of samples in each leaf node, the number of samples per class in each leaf node, and the number of misclassified samples in target leaf nodes, respectively. Binary variable (27) indicates whether leaf node $t$ has the maximum VI index. Finally, $I_{\max}$ (28) represents the maximum VI index value across all target leaf nodes. For a fixed tree depth $D$, the formulation scales linearly in both $|N|$ and $|P|$, but exponentially in $D$, an inherent characteristic shared with Bertsimas & Dunn (2017)'s OCT. Since the new constraints and variables introduced in OPDT add only $O(2^D)$ additional variables

and constraints, which are dominated by the OCT routing constraints, they do not alter the asymptotic complexity. Therefore, OPDT maintains the same $O(2^D \cdot (|P| + |N|))$ model size complexity as OCT.

**Branching Structure Constraints** We incorporate Branching Structure Constraints (BSC) into OPDT to integrate prior knowledge from domain experts. These constraints aim to improve the interpretability and compliance of the resulting rule while enhancing computational efficiency. We design a decision tree that complies with BSC to control: 1) the tree topology through splitting decisions and 2) feature allocation at each branching node through predefined feature groups. Specifically, we define two arrays for branching nodes: $B(split)$, which indicates whether each branching node $i$ needs to be split or not, and $B(group)$, which specifies feature group $G_j$ for branching node $i$, where $G_j$ is a subset of features used for branching. These arrays enable us to construct the desired tree structure and feature allocations over branching nodes to solve the OPDT problem. For leaf nodes, $L(target)$ is a binary array that identifies which leaf nodes are target nodes in the objective function, determined by a domain expert to be aligned with $B(split)$. We assume that feature subgroups for $B(group)$ can be determined by domain expert knowledge. However, this feature grouping is optional as mentioned in Section 3.1. When predefined feature groups are unavailable, OPDT has the flexibility to use a default feature group $G_A$ containing all features. By assigning $G_A$ to all branching nodes, OPDT can enumerate all feature combinations without any subgrouping constraints (e.g., not even requiring separation into numerical and categorical features). Finally, BSC adds additional constraints on $d_t$, $a_{pt}$, $b_t$, and $T_{obj}$ from Algorithm 1 to the main OPDT MIP formulation.

In this paper, we use only the basic numerical-categorical group separation without considering domain-specific knowledge for feature grouping. Specifically, we define $G_N$ as the set of numerical feature groups (i.e., feature groups containing only numerical features) and $G_C$ as the set of categorical feature groups (i.e., feature groups containing only one-hot encoded categorical features). $P_N(g)$ denotes the set of features in numerical feature group $g \in G_N$ and $P_C(g)$ denotes the set of features in categorical feature group $g \in G_C$, where $P_N(g), P_C(g) \subseteq P$. This basic grouping strategy demonstrates that even without sophisticated domain knowledge, OPDT can achieve significant computational gains through feature subgrouping, as we demonstrate in Section 4.

---

**Algorithm 1:** Branching Structure Constraints (BSC)

---

**1 Input:** $|B(\text{split})|, |B(\text{group})|, |L(\text{target})|$
```
// B(split):  binary array to indicate branching splits
// B(group):  subset of features allocated to each branching node
// L(target):  binary array to indicate target leaf nodes
// P_N(g):  set of features in numerical feature group g ∈ G_N
// P_N(g):  set of features in categorical feature group g ∈ G_C
```
**2 foreach** $t \in T_B$ **do**
**3**     $d_t = B(\text{split})[t]$
**4**     $g \leftarrow B(\text{group})[t]$
**5**     **if** $g = G_A$ **then**
**6**        $a_{pt} \geq 0, \qquad \forall p \in P$
**7**     **else if** $g \in G_N$ **then**
**8**        $a_{pt} = 0, \qquad \forall p \in P \setminus P_N(g)$
**9**     **else if** $g \in G_C$ **then**
**10**       $a_{pt} = 0, \qquad \forall p \in P \setminus P_C(g); \ b_t = 0.5$
**11** $T_{obj} = \{t \in T_L | L(\text{target})[t] = 1\}$

---

## 4 Computational Results

In this section, we evaluate the performance of OPDT on datasets from the UCI Machine Learning Repository (Kelly et al., 2017). Since the purpose of this paper is to identify a single pattern in data that maximizes coverage while minimizing the false positive rate due to misclassification, for algorithms that generate

multiple rules, we analyze the best candidate rule among the set of rules generated by ruleset extracting models such as BRS (Wang et al., 2017), IDS (Lakkaraju et al., 2016), IREP (Fürnkranz & Widmer, 1994), PRISM (Cendrowska, 1988), and Ripper (Cohen et al., 1995). All experiments were conducted on an Apple M2 system with an 8-core CPU and 8GB of RAM. In all experiments, we set the weight $w$ for the VI index to 10 as the default value.

## 4.1 Datasets

For our experiments, we use all 15 publicly available datasets from the UCI Machine Learning Repository (Kelly et al., 2017) as detailed in Table 3. For domains that require high interpretability and reliability, we focus on healthcare and finance datasets (Quinlan, 1987; Hofmann, 1994; Yeh, 2008; Wolberg et al., 1995; Rubini et al., 2015; Islam et al., 2020; Salzberg, 1988; Gil & Girela, 2012; Janosi et al., 1988; Chicco & Jurman, 2020; Gong, 1983; Ramana et al., 2012; Little, 2007; Sigillito, 1990; Lubicz et al., 2014). We apply an 80/20 train-test split for our experiments.

Table 3: UCI Dataset

| Dataset | n_instances (n_missing) | n_features (num, cat) | weights (neg, pos) | description |
|---|---|---|---|---|
| Australian | 653 (37) | 15 (6, 9) | (357, 296) | credit approval |
| German | 1000 (0) | 20 (7, 13) | (300, 700) | credit approval |
| Blood Transfusion | 748 (0) | 4 (4, 0) | (570, 178) | healthcare |
| Breast Cancer | 569 (0) | 30 (30, 0) | (212, 357) | healthcare |
| Chronic Kidney | 215 (185) | 24 (11, 13) | (128, 87) | healthcare |
| Early Stage Diabetes | 520 (0) | 16 (1, 15) | (320, 200) | healthcare |
| Echocardiogram | 62 (69) | 7 (4, 3) | (44, 18) | healthcare |
| Fertility | 100 (0) | 9 (2, 7) | (88, 12) | healthcare |
| Heart Disease | 297 (6) | 13 (6, 7) | (160, 137) | healthcare |
| Heart Failure | 299 (0) | 12 (7, 5) | (203, 96) | healthcare |
| Hepatitis | 80 (75) | 19 (6, 13) | (67, 13) | healthcare |
| Indian Liver Patient | 579 (4) | 10 (9, 1) | (414, 165) | healthcare |
| Parkinsons | 195 (0) | 23 (22, 1) | (147, 48) | healthcare |
| Pima Indians Diabetes | 768 (0) | 8 (8, 0) | (500, 268) | healthcare |
| Thoracic Surgery | 470 (0) | 16 (3, 13) | (400, 70) | healthcare |

## 4.2 Performance Enhancements

Generally, solving an ODT is known to be an NP-hard problem (Laurent & Rivest, 1976). To address this computational challenge, we propose three complementary techniques to enhance performance: 1) Branching Structure Constraints (BSC), which leverages prior knowledge about feature relationships to reduce the search space, 2) warmstart initialization, which provides a quality initial solution to accelerate convergence, and 3) branching priority orders, which guides the search process by prioritizing promising decision variables. These techniques work synergistically - BSC narrows the feasible region, warmstart provides a good starting point within this reduced space, and priority ordering helps efficiently navigate toward optimal solutions. Our empirical results demonstrate that this combination significantly reduces computational time while maintaining or improving solution quality across diverse datasets.

First, BSC enhances both interpretability and computational efficiency by imposing meaningful constraints on the branching structure of decision trees. Table 4, with a 10-minute time limit, demonstrates significant computational savings when prior knowledge about rule structure is incorporated. The impact is particularly dramatic for certain rule structures - for instance, employing `{categorical}-{numerical}` constraints reduces runtime from 1232 seconds to 23 seconds for Heart Disease and from 924 seconds to 21 seconds for Thoracic Surgery. Furthermore, decomposing features into numerical and categorical subgroups proves highly effective. In the case of Chronic Kidney Disease, this systematic decomposition strategy reduces the total runtime from 978 seconds (by all features) to approximately 45 seconds (by decomposing features). Beyond runtime improvements, the systematic feature decomposition assists in feature pruning by identifying essential structural patterns for optimality. For example, in Thoracic Surgery, while optimality of the solution with $VI = 42$ remains uncertain due to the time limit being hit on one of feature decomposition,

any potential better solution must exist under the `{numerical}`–`{numerical}` rule structure, as all other feature combinations involving categorical features were solved to optimality within the time limit. This effectively eliminates categorical features from the search space for an optimal solution. Such insights significantly reduce the search space by allowing us to focus computational resources on the most promising feature combinations. In real-world applications, prior domain knowledge could enable even finer feature groupings beyond the basic numerical-categorical group separation. Therefore, BSC not only enhances computational efficiency but also provides valuable guidance toward optimal solutions through strategic feature decomposition.

Table 4: OPDT Runtime (in seconds) and VI by Branching Structure Constraints (BSC)

| | Structure Constraints (solver time limit = 600 sec) | | | | | | | | | |
| | all–all | | num–num | | num–cat | | cat–num | | cat–cat | |
| Dataset | Runtime | VI | Runtime | VI | Runtime | VI | Runtime | VI | Runtime | VI |
|---|---|---|---|---|---|---|---|---|---|---|
| Australian | 600.04 | 143 | 982.97 | 49 | 975.71 | 143 | 45.66 | 143 | 7.29 | 124 |
| German | 1475.81 | 31 | 631.47 | 35 | 1065.43 | 45 | 1006.65 | 45 | 968.56 | 44 |
| Chronic Kidney | 977.94 | 99 | 23.35 | 93 | 16.66 | 99 | 4.64 | 99 | 0.57 | 88 |
| Early Stage Diabetes | 23.07 | 151 | 3.53 | 25 | 6.88 | 130 | 1.32 | 130 | 8.27 | 151 |
| Echocardiogram | 12.08 | 13 | 2.03 | 13 | 0.85 | 13 | 0.19 | 13 | 0.05 | 6 |
| Fertility | 5.33 | 31 | 0.36 | 31 | 0.74 | 30 | 0.23 | 30 | 0.11 | 24 |
| Heart Disease | 1232.83 | 31 | 1061.62 | 26 | 947.29 | 31 | 22.67 | 31 | 1.93 | 17 |
| Heart Failure | 1017.01 | 61 | 1410.82 | 61 | 4.01 | 21 | 79.37 | 21 | - | - |
| Hepatitis | 10.70 | 40 | 1.82 | 40 | 0.50 | 38 | 0.15 | 38 | 0.11 | 35 |
| Indian Liver Patient | 1421.23 | 75 | 600.02 | 75 | 0.41 | 45 | 6.18 | 45 | - | - |
| Parkinsons | 600.02 | 86 | 954.41 | 86 | 4.08 | 64 | 0.96 | 64 | - | - |
| Thoracic Surgery | 924.10 | 39 | 964.28 | 34 | 77.15 | 42 | 20.54 | 42 | 2.34 | 42 |

For warmstart initialization, we develop a novel heuristic algorithm adapted from CART (Breiman et al., 1984) for structure constraints, called Branching Structure Constrained Classification and Regression Tree (BSCCART), as no existing heuristics for structure-constrained decision trees are available. BSCCART modifies CART's branching criteria by restricting feature selection at each node to comply with the branching structure constraints. By applying the same structure constraints used in OPDT, the BSCCART solution can serve as a warmstart for OPDT. The performance comparison between BSCCART and OPDT is presented in Tables 6 and 8. For example, Figure 3 illustrates that the pattern discovered by BSCCART is improved by OPDT on the German dataset. While BSCCART produces a best VI of 8, OPDT improves this to 32 in 12 seconds (highlighted in red), progressing toward the optimal solution. This example demonstrates how OPDT can improve upon a heuristic solution obtained from ML algorithms or prior knowledge. For branching priority orders, we assign higher priorities to topology variables (e.g., $a_{pt}$ and $d_t$), guiding the MIP solver to decide the tree structure variables before others.

Table 5 presents the computational runtime and VI values across different configurations of warmstart and branching priority orders, where `on/on` denotes the use of both strategies. Generally, the `on/on` configuration demonstrates superior or equivalent performance compared to other configurations, showing trends toward higher VI values or faster runtimes. This is particularly evident in datasets solved to optimality within the time limit, such as Early Stage Diabetes, Echocardiogram, Fertility, and Hepatitis, which show identical VI values across all configurations. For these datasets, the runtime analysis clearly demonstrates that both warmstart and branching priority orders contribute to faster computation times compared to their `off` counterparts. Furthermore, the `on/on` configuration exhibits more stable performance with moderate standard deviations, while the `off/off` configuration often shows larger variability. However, the optimal configuration appears to be dataset-dependent, with some exceptions: Blood Transfusion achieves slightly better VI with `on/off`, Indian Liver Patient performs best with `on/off`, and Breast Cancer shows optimal results with `off/on`. Overall, despite some dataset-specific variations, the combined use of warmstart and branching priority orders (`on/on`) enhances both solution quality and computational efficiency across diverse datasets.

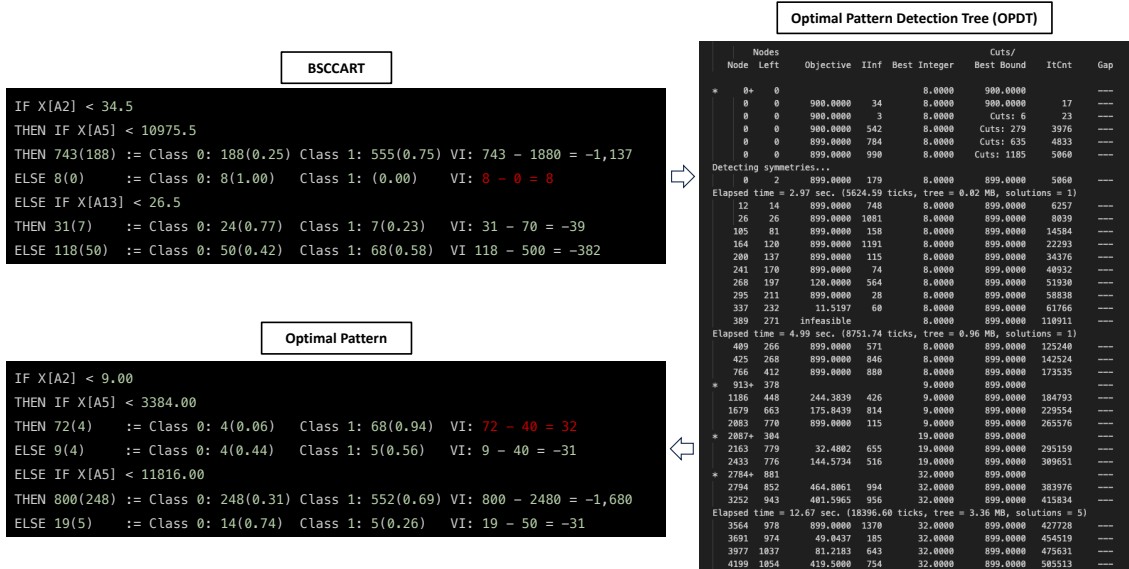

Figure 3: VI improvement by OPDT over BSCCART on the German dataset given $w = 10$. BSCCART, a heuristic method, yields a best VI value of 8. OPDT enhances it to 32 (highlighted in red), achieved in 12 seconds.

Table 5: OPDT Runtime (in seconds) and VI by Warmstart and Branching Priority Orders. **Bold** represents the highest VI value

| | Warmstart from BSCCART / Branching Priority Orders (solver time limit = 120 sec) | | | | | | | |
|---|---|---|---|---|---|---|---|---|
| | on / on | | on / off | | off / on | | off / off | |
| Dataset | Runtime | VI | Runtime | VI | Runtime | VI | Runtime | VI |
| Australian | 18.715 | **134.300** | 26.616 | **134.300** | 23.496 | 132.400 | 54.946 | **134.300** |
| | (±16.804) | (±12.996) | (±33.923) | (±12.996) | (±21.194) | (±12.963) | (±49.832) | (±12.781) |
| German | 29.296 | **53.800** | 11.331 | 51.300 | 42.333 | 33.800 | 108.277 | 24.600 |
| | (±39.142) | (±17.568) | (±20.068) | (±17.468) | (±40.723) | (±12.109) | (±297.212) | (±19.856) |
| Blood Transfusion | 7.574 | 51.000 | 18.458 | **51.200** | 9.565 | 50.100 | 6.575 | 50.800 |
| | (±9.662) | (±10.509) | (±26.890) | (±10.443) | (±12.220) | (±12.387) | (±6.100) | (±10.696) |
| Breast Cancer | 37.938 | 230.100 | 29.135 | 230.000 | 42.666 | **231.000** | 68.280 | 223.300 |
| | (±30.127) | (±5.782) | (±19.873) | (±5.774) | (±32.837) | (±7.630) | (±26.746) | (±16.958) |
| Chronic Kidney | 1.230 | 98.700 | 3.675 | **99.200** | 3.201 | **99.200** | 4.746 | 98.500 |
| | (±1.101) | (±3.234) | (±3.503) | (±3.327) | (±2.730) | (±3.327) | (±7.048) | (±3.408) |
| Early Stage Diabetes | 1.302 | **153.400** | 1.811 | **153.400** | 3.230 | **153.400** | 4.536 | **153.400** |
| | (±1.032) | (±3.565) | (±1.275) | (±3.565) | (±1.231) | (±3.565) | (±1.904) | (±3.565) |
| Echocardiogram | 0.162 | **18.667** | 0.233 | **18.667** | 0.664 | **18.667** | 0.654 | **18.667** |
| | (±0.180) | (±4.743) | (±0.263) | (±4.743) | (±0.552) | (±4.743) | (±0.563) | (±4.743) |
| Fertility | 0.073 | **34.300** | 0.101 | **34.300** | 0.252 | **34.300** | 0.342 | **34.300** |
| | (±0.089) | (±1.947) | (±0.141) | (±1.947) | (±0.118) | (±1.947) | (±0.164) | (±1.947) |
| Heart Disease | 14.438 | **28.200** | 18.029 | 26.700 | 9.782 | 26.100 | 15.690 | 27.700 |
| | (±19.564) | (±4.709) | (±29.863) | (±4.423) | (±12.683) | (±3.178) | (±16.870) | (±4.832) |
| Heart Failure | 10.971 | **62.100** | 11.723 | 61.800 | 11.219 | 60.200 | 16.533 | 61.500 |
| | (±17.315) | (±5.859) | (±19.361) | (±5.996) | (±11.942) | (±4.467) | (±14.937) | (±6.329) |
| Hepatitis | 0.667 | **42.200** | 0.869 | **42.200** | 0.653 | **42.200** | 0.703 | **42.200** |
| | (±1.357) | (±3.393) | (±1.834) | (±3.393) | (±0.551) | (±3.393) | (±0.428) | (±3.393) |
| Indian Liver Patient | 14.701 | 82.300 | 21.831 | **84.000** | 27.622 | 83.800 | 23.099 | 76.200 |
| | (±15.843) | (±8.220) | (±21.734) | (±7.454) | (±31.831) | (±7.361) | (±28.159) | (±15.817) |
| Parkinsons | 25.639 | **82.700** | 15.436 | 82.500 | 30.355 | **82.700** | 15.611 | **82.700** |
| | (±34.000) | (±3.268) | (±23.398) | (±3.308) | (±32.283) | (±3.268) | (±15.215) | (±3.268) |
| Pima Indians Diabetes | 22.204 | **91.200** | 26.598 | 90.900 | 21.091 | 91.100 | 54.652 | 89.500 |
| | (±33.173) | (±5.329) | (±20.149) | (±6.173) | (±23.323) | (±5.322) | (±30.207) | (±6.704) |
| Thoracic Surgery | 29.033 | **46.200** | 21.901 | 45.600 | 26.106 | 45.444 | 32.679 | **46.200** |
| | (±28.946) | (±5.095) | (±14.735) | (±4.812) | (±27.370) | (±5.223) | (±20.418) | (±5.095) |

### 4.3 Computational Experiments

In this paper, we preprocess features differently according to each algorithm's requirements. For RSCRULES, IREP, PRISM, IDS, and RIPPER, we discretize numerical features into ten equal bins. For BRS, we binarize features into a binary format. For OPDT and BSCRULES, while continuous numerical features can be used without transformation, categorical features must be one-hot encoded. Regarding rule complexity, we limit the length of rule conditions to two for RIPPER, BRS in ruleset extracting models, and all structure-constrained models as shown in Tables 6 and 8. Other algorithms, having no such rule length parameter, can generate rules with conditions longer than two as they have more flexibility in rule generation. To provide another structure-constrained method alongside BSCCART, we develop a simple rule structure-constrained (RSC) algorithm based on Rules Extraction System (RULES) (Pham & Aksoy, 1995), which we call RSCRULES. Finally, we impose a 10-minute time limit on OPDT runtime.

For experiments, we create 10 random data splits through shuffling. For each split, we compute performance metrics and report their averages across the splits. Additionally, since RIPPER and BRS are non-deterministic algorithms, we run them with multiple random seeds on each split: 50 different seeds for BRS and 10 different seeds for RIPPER. The computational details are provided in Table 10 in the Appendix. We compare the induced rules based on their precision, coverage, and volume-impurity. Precision is calculated as the ratio between the number of samples correctly classified by a rule and the total number of samples covered by the rule. Coverage is computed as the ratio between the number of samples covered by the rule and the total number of samples in the dataset. VI is computed based on volume and misclassified samples with respect to weight $w = 10$. We use test data to evaluate how well the induced rule captures a common pattern in unseen data.

Table 6: OPDT Performance Compared to Benchmark Methods on Training Datasets (80%). **Bold** indicates ranking in top 4 methods, and underlined represents the best performance across all methods.

| Dataset | | Training Data (80%) | | | | | | | |
| --- | --- | --- | --- | --- | --- | --- | --- | --- | --- |
| | | Structure-constrained Models | | | Ruleset Extracting Models | | | | |
| | | OPDT | BSCCART | RSCRULES | BRS | IDS | IREP | PRISM | Ripper |
| Australian | VI | **135.60** | **130.50** | 103.40 | **116.72** | 36.30 | 103.40 | 44.20 | **114.38** |
| | Time | 602.00 | 0.25 | 0.04 | 0.36 | 0.22 | 0.08 | 3.03 | 2.01 |
| German | VI | **60.00** | **49.80** | **57.80** | 16.27 | 19.20 | -32.20 | 20.10 | **47.62** |
| | Time | 749.82 | 0.20 | 0.10 | 0.53 | 0.25 | 0.09 | 7.20 | 0.17 |
| Blood Transfusion | VI | **51.20** | **45.50** | 24.00 | -583.40 | 12.20 | 10.72 | 15.10 | **18.17** |
| | Time | 90.70 | 0.13 | 0.02 | 0.23 | 0.23 | 0.04 | 0.53 | 0.07 |
| Breast Cancer | VI | **230.00** | **183.50** | 124.60 | 114.00 | 111.70 | 50.51 | **114.30** | 50.50 |
| | Time | 790.97 | 7.25 | 0.06 | 0.59 | 0.22 | 0.39 | 2.95 | 0.49 |
| Chronic Kidney | VI | **99.20** | **98.10** | 52.90 | 46.31 | 49.60 | **79.10** | 52.90 | **84.00** |
| | Time | 6.60 | 0.24 | 0.02 | 0.33 | 0.19 | 0.13 | 0.29 | 0.15 |
| Early Stage Diabetes | VI | **153.40** | **153.40** | 121.20 | **151.83** | 119.30 | 118.16 | 90.60 | **151.75** |
| | Time | 3.71 | 0.03 | 0.02 | 0.54 | 0.21 | 0.04 | 0.63 | 0.08 |
| Echocardiogram | VI | **18.10** | **14.50** | **9.90** | -8.13 | **7.50** | 4.31 | 7.10 | 5.74 |
| | Time | 0.80 | 0.07 | 0.01 | 0.24 | 0.18 | 0.05 | 0.18 | 0.07 |
| Fertility | VI | **34.30** | **34.30** | 17.40 | **18.71** | **20.30** | 16.19 | 15.00 | 17.18 |
| | Time | 0.25 | 0.02 | 0.01 | 0.27 | 0.19 | 0.03 | 0.13 | 0.04 |
| Heart Disease | VI | **29.20** | **16.80** | **20.50** | 15.01 | **15.10** | -5.89 | 13.30 | 13.59 |
| | Time | 524.81 | 0.14 | 0.03 | 0.36 | 0.20 | 0.06 | 1.01 | 0.10 |
| Heart Failure | VI | **62.10** | **29.60** | 16.30 | -41.97 | 19.40 | 17.58 | **20.70** | **21.41** |
| | Time | 729.32 | 0.36 | 0.03 | 0.30 | 0.19 | 0.07 | 0.82 | 0.11 |
| Hepatitis | VI | **42.20** | **40.10** | 17.50 | **29.31** | 25.00 | 26.33 | 21.80 | **28.00** |
| | Time | 0.82 | 0.10 | 0.01 | 0.27 | 0.19 | 0.06 | 0.15 | 0.07 |
| Indian Liver Patient | VI | **82.50** | **54.50** | 19.60 | -790.93 | 17.40 | **42.62** | 18.10 | **46.09** |
| | Time | 999.70 | 0.61 | 0.04 | 0.27 | 0.21 | 0.09 | 2.98 | 0.14 |
| Parkinsons | VI | **82.00** | **70.20** | 28.50 | 52.25 | 25.40 | 17.56 | 26.70 | 18.16 |
| | Time | 863.11 | 1.77 | 0.03 | 0.37 | 0.19 | 0.35 | 1.56 | 0.41 |
| Pima Indians Diabetes | VI | **89.10** | **36.20** | 37.40 | -920.73 | 21.50 | 22.69 | 22.80 | **26.88** |
| | Time | 1047.67 | 0.76 | 0.07 | 0.35 | 0.22 | 0.08 | 3.35 | 0.13 |
| Thoracic Surgery | VI | **46.20** | **20.20** | **31.70** | -21.07 | 14.40 | -2.14 | 16.90 | **26.36** |
| | Time | 133.21 | 0.10 | 0.02 | 2.13 | 0.21 | 0.05 | 1.70 | 2.16 |

First, Table 6 shows that OPDT can find better common patterns in data compared to other models, including ruleset extracting models that have more flexibility in rule length. Regarding runtime in Table 10 in the Appendix, we measure total runtime including warmstart generation, model initialization, and 10 minutes of solver time, which can exceed 600 seconds. In other words, a runtime less than 600 seconds indicates the problem was solved to optimality. Table 10 demonstrates that several datasets were solved within the time limit, showing that OPDT is applicable for finding an optimal rule in moderate-sized datasets. Furthermore, as shown in Table 7, even for datasets that do not reach optimality within the 10-minute limit, the best solutions are typically found within the first 2 minutes. Second, among the structure-constrained heuristic methods we developed, Table 6 shows that BSCCART performs better than RSCRULES. Moreover, BSCCART achieves the second-best performance in finding high-quality rules among all methods while complying with structure constraints given by domain experts. Among the ruleset extracting models, RIPPER shows comparably good performance to other methods. As shown in Table 10, OPDT requires more than 10 minutes to prove optimality for some datasets. This could raise concerns about the scalability of OPDT to real-world applications. However, in this paper, we use only the basic numerical-categorical group separation without considering domain-specific knowledge for feature grouping. As shown in Table 4, if feature grouping is actively utilized, computational efficiency can be enhanced significantly. We further evaluate the induced rules from each model on test datasets. Table 8 shows that OPDT achieves the best VI scores in 6 datasets, which is double that of BSCCART or RIPPER that have the best VI in 3 datasets. Furthermore, OPDT performs within the top 4 in most datasets, with exceptions in 5 datasets. BSCCART and RIPPER maintain their strong performance on test datasets, consistent with their performance in training datasets, showing the second-best results after OPDT.

However, it is worth noting that OPDT's strong training performance does not always transfer to the test set. For example, even though OPDT achieves a higher VI value than BSCCART on training data for the German and Heart Disease datasets, it fails to show better VI on test data. Specifically, in the German dataset, OPDT achieves precision 0.936 and coverage 0.225 on training data, but precision 0.883 and coverage 0.231 on test data, resulting in a test VI of $-7.9$. In contrast, BSCCART achieves precision 0.920 and coverage 0.315 on training data, and precision 0.896 and coverage 0.326 on test data, resulting in a test VI of $-1.8$. A similar pattern is observed in the Heart Disease dataset: OPDT achieves precision 0.979 and coverage 0.165 on training data, but precision 0.800 and coverage 0.180 on test data, yielding a test VI of $-9.2$, whereas BSCCART achieves precision 0.926 and coverage 0.269 on training data and precision 0.944 and coverage 0.250 on test data, yielding a test VI of 5.0. In both cases, OPDT identifies a narrower region (lower coverage than BSCCART on training data) to achieve higher precision. However, this precision advantage is not stable on the test set. This suggests that the weight parameter $w$ may force OPDT toward a narrow region that fails to represent the whole dataset, implying that $w = 10$ may not be an appropriate value for representing the dominant pattern in these datasets. As another case, in the Thoracic Surgery dataset, while BSCCART, IDS, and PRISM identify rules with very narrow coverage (0.064, 0.038, and 0.045, respectively) but near-perfect precision (0.991, 1.000, and 1.000) on training data, OPDT identifies a substantially broader rule with coverage 0.264 and precision 0.949, resulting in a higher training VI of 46.2 compared to 20.2, 14.4, and 16.9 for BSCCART, IDS, and PRISM, respectively. However, on test data, OPDT's broader rule yields a test VI of $-9.8$, whereas BSCCART, IDS, and PRISM achieve better test VI values of $-3.8$, $-0.1$, and 4.5, respectively. Even though OPDT discovers a more representative pattern in data with broader coverage (0.247) and reasonable precision (0.855) than BSCCART (precision 0.521, coverage 0.066), IDS (precision 0.933, coverage 0.031), and PRISM (precision 1.000, coverage 0.048), the weight parameter $w$ penalizes OPDT's broader coverage rule. This further supports the observation that the choice of $w$ is critical to finding a representative pattern in data. Additionally, it is worth noting that IDS and PRISM are not constrained to rules with only two conditions, unlike OPDT, potentially giving them an advantage in searching for patterns having more than two conditions.

Finally, Table 9 presents the sensitivity analysis of the weight parameter $w$ on the UCI Hepatitis dataset. As $w$ decreases from 10 to 2, OPDT exhibits a monotone precision–coverage trade-off, yielding higher coverage at the cost of reduced precision, confirming that $w$ functions as an adjustable precision requirement. Runtime remains stable across all $w$ values, and test performance closely tracks training performance, indicating strong out-of-sample performance.

Table 7: OPDT Runtime and Time to Solution (in seconds) without BSC. Runtime represents the total computational time including optimality proof, while Time to Solution represents when the final improvement to the best solution was made.

| Dataset | Runtime | Time to Solution |
|---|---|---|
| Australian | 139.363 (±45.693) | 11.625 (±14.371) |
| German | 930.113 (±382.954) | 104.823 (±141.195) |
| Blood Transfusion | 776.287 (±534.019) | 2.771 (±2.539) |
| Breast Cancer | 1096.570 (±234.938) | 125.454 (±300.844) |
| Chronic Kidney | 916.377 (±292.314) | 119.097 (±169.979) |
| Early Stage Diabetes | 112.018 (±311.394) | 2.764 (±2.390) |
| Echocardiogram | 109.457 (±321.932) | 0.176 (±0.487) |
| Fertility | 4.687 (±1.996) | 0.213 (±0.448) |
| Heart Disease | 904.345 (±380.713) | 41.906 (±85.544) |
| Heart Failure | 187.302 (±42.433) | 17.734 (±27.696) |
| Hepatitis | 8.601 (±3.545) | 0.560 (±1.071) |
| Indian Liver Patient | 1161.727 (±196.850) | 10.659 (±12.828) |
| Parkinsons | 899.464 (±338.915) | 14.228 (±29.471) |
| Pima Indians Diabetes | 600.030 (±0.007) | 140.321 (±182.179) |
| Thoracic Surgery | 585.058 (±47.319) | 28.280 (±32.392) |

Table 8: OPDT Performance Compared to Benchmark Methods on Testing Datasets (20%). **Bold** indicates ranking in top 4 methods, and underlined represents the best performance across all methods.

| | Test Data (20%) | | | | | | | |
|---|---|---|---|---|---|---|---|---|
| | Structure-constrained Models | | | Ruleset Extracting Models | | | | |
| Dataset | OPDT | BSCCART | RSCRULES | BRS | IDS | IREP | PRISM | Ripper |
| Australian | **22.80** | **21.70** | **20.60** | **20.93** | 5.00 | -41.30 | 8.20 | -8.26 |
| German | -7.90 | -1.80 | -4.20 | **-0.71** | **1.80** | -37.32 | **0.80** | **10.52** |
| Blood Transfusion | **-6.50** | **-6.60** | **-6.30** | -299.24 | -13.30 | -28.31 | **-10.10** | -31.68 |
| Breast Cancer | **48.20** | **38.60** | 25.60 | 21.22 | 21.30 | 10.04 | **25.30** | 10.62 |
| Chronic Kidney | **16.80** | **17.60** | 13.10 | 9.33 | 10.10 | **19.83** | 13.10 | **21.80** |
| Early Stage Diabetes | **38.10** | **39.60** | 31.80 | **38.31** | 27.20 | 26.44 | 24.00 | **38.74** |
| Echocardiogram | **-2.20** | **-1.60** | -2.80 | -8.98 | **-2.20** | **-0.10** | -2.30 | **-1.33** |
| Fertility | **6.60** | **6.60** | 0.10 | 1.03 | 3.10 | **3.58** | 3.80 | -1.94 |
| Heart Disease | -9.20 | **5.00** | **-0.90** | **0.64** | **2.10** | -19.49 | 0.20 | -4.52 |
| Heart Failure | **5.10** | **-1.20** | -5.20 | -19.25 | **3.80** | -8.55 | **1.00** | -6.73 |
| Hepatitis | 0.50 | -2.20 | -5.40 | **3.43** | 1.10 | **4.55** | 2.70 | **5.66** |
| Indian Liver Patient | **12.70** | 2.30 | 4.20 | -198.45 | 3.50 | **7.05** | 4.40 | -7.92 |
| Parkinsons | 2.80 | -5.80 | **8.60** | **9.06** | 0.00 | **4.06** | 3.20 | **4.60** |
| Pima Indians Diabetes | **4.90** | -8.40 | **4.90** | -237.90 | **3.70** | -9.60 | **1.60** | -8.96 |
| Thoracic Surgery | -9.80 | **-3.80** | **-4.90** | -27.67 | **-0.10** | -12.72 | **4.50** | -8.17 |

Table 9: Sensitivity analysis of the weight $w$ on the UCI Hepatitis dataset (Training: 64, Test: 16). For precision, the counts in parentheses denote the number of true positives out of the total fired samples (e.g., 0.978 (45/46) denotes 45 true positives out of 46 fired samples). For coverage, the counts in parentheses denote the total fired samples out of the total samples in the dataset (e.g., 0.719 (46/64) denotes 46 fired samples out of 64 total samples). VI denotes the VI value in training data and Runtime denotes the total computational time in seconds.

| | Training Data (80%) | | Test Data (20%) | | | |
|---|---|---|---|---|---|---|
| $w$ | Precision | Coverage | Precision | Coverage | VI | Runtime |
| 10 | 1.000 (38/38) | 0.594 (38/64) | 1.000 (9/9) | 0.562 (9/16) | 38 | 13.83 |
| 9 | 1.000 (38/38) | 0.594 (38/64) | 1.000 (9/9) | 0.562 (9/16) | 38 | 11.45 |
| 8 | 0.978 (45/46) | 0.719 (46/64) | 0.923 (12/13) | 0.812 (13/16) | 38 | 9.62 |
| 7 | 0.978 (45/46) | 0.719 (46/64) | 0.923 (12/13) | 0.812 (13/16) | 39 | 8.92 |
| 6 | 0.978 (45/46) | 0.719 (46/64) | 0.923 (12/13) | 0.812 (13/16) | 40 | 10.25 |
| 5 | 0.978 (45/46) | 0.719 (46/64) | 0.923 (12/13) | 0.812 (13/16) | 41 | 8.34 |
| 4 | 0.960 (48/50) | 0.781 (50/64) | 0.923 (12/13) | 0.812 (13/16) | 42 | 9.45 |

Table 9 continued from previous page

| | Train | | Test | | | |
|---|---|---|---|---|---|---|
| $w$ | Precision | Coverage | Precision | Coverage | VI | Runtime |
| 3 | 0.960 (48/50) | 0.781 (50/64) | 0.929 (13/14) | 0.875 (14/16) | 44 | 7.06 |
| 2 | 0.898 (53/59) | 0.922 (59/64) | 0.875 (14/16) | 1.000 (16/16) | 47 | 7.74 |

## 5 Conclusion and Future Work

In this paper, we introduce an RBML model that offers high interpretability and compliance for rule discovery. Our proposed model, OPDT, is based on an optimal decision tree, incorporating customizable rule structures based on a decision maker's needs. OPDT effectively identifies and describes significant patterns within a reasonable time while adhering to desired rule structures and precision requirements. To enhance computational efficiency, we develop three key techniques: Branching Structure Constraints (BSC), warmstart initialization with BSCCART, and branching priority orders. Our experimental results across 15 UCI datasets demonstrate that OPDT consistently finds high-quality rules that comply with given structure constraints. Moreover, the systematic feature decomposition approach not only improves computational efficiency but also provides valuable guidance toward optimal solutions through strategic feature grouping. This work contributes to rule-based machine learning by providing a flexible framework for discovering interpretable rules that balance solution quality, structure compliance, and computational time.

Despite the contributions presented in this paper, a number of important challenges remain to be addressed in future work. First, even though OPDT can find an optimal pattern given a specific rule structure and weight, it requires a decision maker to specify these parameters appropriately to identify a representative pattern in data that generalizes beyond the training data. Second, while OPDT is designed to extract a single optimal pattern, it is natural to extend it to discover a set of rules that represent data optimally, which remains an important direction for future work. Finally, by the inherent nature of optimization-based methods, OPDT still has a scalability limitation on large-scale data. As demonstrated by the speedup achieved through BSC with different feature groupings, further research is needed to develop systematic approaches for utilizing the BSC framework with prior knowledge tailored to specific datasets.

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

# Appendices

# A    OPDT and Benchmark Methods Performance

Table 10: OPDT and Benchmark Methods Performance on Training Data (80%) and Test Data (20%): mean (± standard deviation) across 10 random data splits

| Dataset | Runtime | Precision | Coverage | VI | Precision | Coverage | VI |
|---|---|---|---|---|---|---|---|
| | | Training Data (80%) | | | Test Data (20%) | | |
| **Australian** | | | | | | | |
| - OPDT | 602.00 | 0.963 | 0.411 | 135.600 | 0.940 | 0.426 | 22.800 |
| | (±482.907) | (±0.007) | (±0.018) | (±12.492) | (±0.028) | (±0.039) | (±15.091) |
| - BSCCART | 0.25 | 0.957 | 0.442 | 130.500 | 0.937 | 0.456 | 21.700 |
| | (±0.175) | (±0.007) | (±0.015) | (±14.254) | (±0.023) | (±0.040) | (±13.598) |
| - RSCRULES | 0.04 | 0.943 | 0.462 | 103.400 | 0.933 | 0.478 | 20.600 |
| | (±0.011) | (±0.006) | (±0.006) | (±14.010) | (±0.023) | (±0.026) | (±14.010) |
| - BRS | 0.36 | 0.950 | 0.453 | 116.720 | 0.934 | 0.470 | 20.930 |
| | (±0.009) | (±0.006) | (±0.020) | (±13.667) | (±0.022) | (±0.029) | (±13.623) |
| - IDS | 0.22 | 1.000 | 0.070 | 36.300 | 0.955 | 0.069 | 5.000 |
| | (±0.005) | (±0.000) | (±0.028) | (±14.863) | (±0.065) | (±0.032) | (±5.598) |
| - IREP | 0.08 | 0.943 | 0.462 | 103.400 | 0.811 | 0.315 | -41.303 |
| | (±0.001) | (±0.006) | (±0.006) | (±14.010) | (±0.043) | (±0.089) | (±27.431) |
| - PRISM | 3.03 | 1.000 | 0.085 | 44.200 | 0.963 | 0.085 | 8.200 |
| | (±0.133) | (±0.000) | (±0.007) | (±3.584) | (±0.078) | (±0.025) | (±8.135) |
| - Ripper | 2.01 | 0.952 | 0.428 | 114.384 | 0.826 | 0.106 | -8.260 |
| | (±5.924) | (±0.006) | (±0.014) | (±13.649) | (±0.063) | (±0.051) | (±3.715) |
| **German** | | | | | | | |
| - OPDT | 749.82 | 0.936 | 0.225 | 60.000 | 0.883 | 0.231 | -7.900 |
| | (±370.714) | (±0.011) | (±0.085) | (±12.481) | (±0.020) | (±0.098) | (±10.609) |
| - BSCCART | 0.20 | 0.920 | 0.315 | 49.800 | 0.896 | 0.326 | -1.800 |
| | (±0.126) | (±0.008) | (±0.023) | (±18.920) | (±0.027) | (±0.043) | (±16.498) |
| - RSCRULES | 0.10 | 0.938 | 0.205 | 57.800 | 0.895 | 0.199 | -4.200 |
| | (±0.006) | (±0.012) | (±0.087) | (±13.815) | (±0.032) | (±0.094) | (±10.696) |
| - BRS | 0.53 | 0.908 | 0.349 | 16.266 | 0.901 | 0.361 | -0.712 |
| | (±0.064) | (±0.010) | (±0.017) | (±33.298) | (±0.029) | (±0.030) | (±20.104) |
| - IDS | 0.25 | 1.000 | 0.024 | 19.200 | 0.950 | 0.019 | 1.800 |
| | (±0.009) | (±0.000) | (±0.004) | (±3.360) | (±0.112) | (±0.007) | (±4.077) |
| - IREP | 0.09 | 0.867 | 0.209 | -32.202 | 0.801 | 0.266 | -37.322 |
| | (±0.002) | (±0.014) | (±0.043) | (±13.316) | (±0.023) | (±0.051) | (±13.754) |
| - PRISM | 7.20 | 1.000 | 0.025 | 20.100 | 0.949 | 0.024 | 0.800 |
| | (±0.112) | (±0.000) | (±0.003) | (±2.079) | (±0.086) | (±0.013) | (±4.803) |
| - Ripper | 0.17 | 0.925 | 0.256 | 47.618 | 0.917 | 0.207 | 10.524 |
| | (±0.001) | (±0.010) | (±0.045) | (±17.287) | (±0.017) | (±0.095) | (±8.096) |
| **Blood Transfusion** | | | | | | | |
| - OPDT | 90.70 | 0.948 | 0.220 | 51.200 | 0.892 | 0.220 | -6.500 |
| | (±11.522) | (±0.014) | (±0.055) | (±10.443) | (±0.017) | (±0.072) | (±14.222) |
| - BSCCART | 0.13 | 0.955 | 0.195 | 45.500 | 0.889 | 0.195 | -6.600 |
| | (±0.004) | (±0.023) | (±0.085) | (±10.886) | (±0.025) | (±0.095) | (±15.079) |
| - RSCRULES | 0.02 | 0.963 | 0.088 | 24.000 | 0.841 | 0.087 | -6.300 |
| | (±0.006) | (±0.021) | (±0.044) | (±5.416) | (±0.125) | (±0.053) | (±11.046) |
| - BRS | 0.23 | 0.775 | 0.922 | -583.396 | 0.771 | 0.921 | -299.236 |
| | (±0.042) | (±0.043) | (±0.142) | (±185.327) | (±0.044) | (±0.151) | (±94.622) |
| - IDS | 0.23 | 1.000 | 0.024 | 12.200 | 0.552 | 0.023 | -13.300 |
| | (±0.006) | (±0.000) | (±0.007) | (±3.676) | (±0.337) | (±0.012) | (±11.605) |
| - IREP | 0.04 | 0.931 | 0.085 | 10.720 | 0.785 | 0.124 | -28.306 |
| | (±0.001) | (±0.025) | (±0.012) | (±11.740) | (±0.040) | (±0.018) | (±10.268) |
| - PRISM | 0.53 | 0.993 | 0.034 | 15.100 | 0.707 | 0.036 | -10.100 |
| | (±0.032) | (±0.015) | (±0.013) | (±3.900) | (±0.263) | (±0.019) | (±4.977) |
| - Ripper | 0.07 | 0.959 | 0.071 | 18.166 | 0.687 | 0.112 | -31.678 |
| | (±0.002) | (±0.022) | (±0.023) | (±8.388) | (±0.039) | (±0.054) | (±14.771) |
| **Breast Cancer** | | | | | | | |
| - OPDT | 790.97 | 0.992 | 0.549 | 230.000 | 0.979 | 0.537 | 48.200 |
| | (±255.965) | (±0.004) | (±0.013) | (±8.055) | (±0.018) | (±0.021) | (±10.185) |
| - BSCCART | 7.25 | 0.966 | 0.617 | 183.500 | 0.958 | 0.602 | 38.600 |
| | (±0.105) | (±0.014) | (±0.028) | (±29.289) | (±0.026) | (±0.041) | (±16.460) |

Table 10 continued from previous page

| Dataset | Training Data (80%) | | | | Test Data (20%) | | |
|---|---|---|---|---|---|---|---|
| | Runtime | Precision | Coverage | VI | Precision | Coverage | VI |
| - RSCRULES | 0.06 | 0.971 | 0.386 | 124.600 | 0.962 | 0.365 | 25.600 |
| | (±0.003) | (±0.007) | (±0.029) | (±20.359) | (±0.024) | (±0.043) | (±9.698) |
| - BRS | 0.59 | 1.000 | 0.251 | 113.998 | 0.982 | 0.235 | 21.216 |
| | (±0.092) | (±0.000) | (±0.025) | (±11.555) | (±0.022) | (±0.045) | (±5.170) |
| - IDS | 0.22 | 1.000 | 0.245 | 111.700 | 0.982 | 0.231 | 21.300 |
| | (±0.027) | (±0.000) | (±0.037) | (±16.747) | (±0.031) | (±0.048) | (±7.973) |
| - IREP | 0.39 | 1.000 | 0.111 | 50.514 | 0.975 | 0.110 | 10.041 |
| | (±0.001) | (±0.000) | (±0.007) | (±3.355) | (±0.030) | (±0.014) | (±4.096) |
| - PRISM | 2.95 | 1.000 | 0.251 | 114.300 | 0.995 | 0.239 | 25.300 |
| | (±0.169) | (±0.000) | (±0.024) | (±10.740) | (±0.017) | (±0.045) | (±4.762) |
| - Ripper | 0.49 | 1.000 | 0.111 | 50.502 | 0.993 | 0.100 | 10.618 |
| | (±0.003) | (±0.000) | (±0.007) | (±3.309) | (±0.008) | (±0.023) | (±3.265) |
| Chronic Kidney | | | | | | | |
| - OPDT | 6.60 | 1.000 | 0.577 | 99.200 | 0.966 | 0.600 | 16.800 |
| | (±10.083) | (±0.000) | (±0.019) | (±3.327) | (±0.021) | (±0.041) | (±4.237) |
| - BSCCART | 0.24 | 0.996 | 0.594 | 98.100 | 0.967 | 0.619 | 17.600 |
| | (±0.023) | (±0.005) | (±0.006) | (±4.433) | (±0.030) | (±0.025) | (±7.947) |
| - RSCRULES | 0.02 | 1.000 | 0.308 | 52.900 | 1.000 | 0.305 | 13.100 |
| | (±0.000) | (±0.000) | (±0.009) | (±1.524) | (±0.000) | (±0.035) | (±1.524) |
| - BRS | 0.33 | 0.999 | 0.276 | 46.305 | 0.982 | 0.276 | 9.332 |
| | (±0.066) | (±0.002) | (±0.014) | (±0.964) | (±0.008) | (±0.009) | (±0.879) |
| - IDS | 0.19 | 1.000 | 0.288 | 49.600 | 0.983 | 0.281 | 10.100 |
| | (±0.003) | (±0.000) | (±0.017) | (±2.914) | (±0.053) | (±0.040) | (±6.590) |
| - IREP | 0.13 | 0.984 | 0.551 | 79.096 | 0.996 | 0.488 | 19.834 |
| | (±0.001) | (±0.003) | (±0.012) | (±3.668) | (±0.004) | (±0.046) | (±1.621) |
| - PRISM | 0.29 | 1.000 | 0.308 | 52.900 | 1.000 | 0.305 | 13.100 |
| | (±0.135) | (±0.000) | (±0.009) | (±1.524) | (±0.000) | (±0.035) | (±1.524) |
| - Ripper | 0.15 | 0.987 | 0.563 | 84.004 | 0.998 | 0.520 | 21.796 |
| | (±0.001) | (±0.002) | (±0.006) | (±2.406) | (±0.002) | (±0.044) | (±1.456) |
| Early Stage Diabetes | | | | | | | |
| - OPDT | 3.71 | 0.999 | 0.374 | 153.400 | 0.996 | 0.386 | 38.100 |
| | (±3.314) | (±0.004) | (±0.012) | (±3.565) | (±0.012) | (±0.045) | (±3.725) |
| - BSCCART | 0.03 | 1.000 | 0.369 | 153.400 | 1.000 | 0.381 | 39.600 |
| | (±0.000) | (±0.000) | (±0.009) | (±3.565) | (±0.000) | (±0.034) | (±3.565) |
| - RSCRULES | 0.02 | 0.965 | 0.448 | 121.200 | 0.968 | 0.450 | 31.800 |
| | (±0.009) | (±0.006) | (±0.008) | (±10.163) | (±0.023) | (±0.032) | (±10.163) |
| - BRS | 0.54 | 1.000 | 0.365 | 151.830 | 1.000 | 0.369 | 38.314 |
| | (±0.086) | (±0.000) | (±0.009) | (±3.751) | (±0.001) | (±0.033) | (±3.537) |
| - IDS | 0.21 | 1.000 | 0.287 | 119.300 | 0.995 | 0.271 | 27.200 |
| | (±0.003) | (±0.000) | (±0.067) | (±27.745) | (±0.015) | (±0.082) | (±9.953) |
| - IREP | 0.04 | 0.966 | 0.437 | 118.158 | 0.970 | 0.381 | 26.444 |
| | (±0.001) | (±0.006) | (±0.019) | (±12.140) | (±0.017) | (±0.063) | (±7.322) |
| - PRISM | 0.63 | 1.000 | 0.218 | 90.600 | 1.000 | 0.231 | 24.000 |
| | (±0.059) | (±0.000) | (±0.056) | (±23.201) | (±0.000) | (±0.062) | (±6.446) |
| - Ripper | 0.08 | 0.999 | 0.371 | 151.746 | 0.999 | 0.375 | 38.740 |
| | (±0.002) | (±0.001) | (±0.010) | (±3.258) | (±0.001) | (±0.033) | (±3.551) |
| Echocardiogram | | | | | | | |
| - OPDT | 0.80 | 0.988 | 0.430 | 18.100 | 0.762 | 0.369 | -2.200 |
| | (±0.618) | (±0.020) | (±0.094) | (±4.818) | (±0.317) | (±0.191) | (±6.477) |
| - BSCCART | 0.07 | 0.961 | 0.500 | 14.500 | 0.886 | 0.492 | -1.600 |
| | (±0.013) | (±0.036) | (±0.040) | (±7.990) | (±0.136) | (±0.126) | (±9.477) |
| - RSCRULES | 0.01 | 0.986 | 0.263 | 9.900 | 0.598 | 0.246 | -2.800 |
| | (±0.001) | (±0.022) | (±0.163) | (±4.886) | (±0.439) | (±0.248) | (±5.138) |
| - BRS | 0.24 | 0.937 | 0.429 | -8.132 | 0.782 | 0.428 | -8.982 |
| | (±0.035) | (±0.041) | (±0.118) | (±15.676) | (±0.110) | (±0.174) | (±3.871) |
| - IDS | 0.18 | 1.000 | 0.153 | 7.500 | 0.831 | 0.215 | -2.200 |
| | (±0.003) | (±0.000) | (±0.036) | (±1.780) | (±0.171) | (±0.149) | (±3.994) |
| - IREP | 0.05 | 0.949 | 0.267 | 4.307 | 0.935 | 0.229 | -0.102 |
| | (±0.001) | (±0.032) | (±0.093) | (±5.100) | (±0.087) | (±0.095) | (±3.682) |
| - PRISM | 0.18 | 1.000 | 0.145 | 7.100 | 0.500 | 0.131 | -2.300 |
| | (±0.016) | (±0.000) | (±0.095) | (±4.654) | (±0.456) | (±0.195) | (±4.644) |
| - Ripper | 0.07 | 0.966 | 0.223 | 5.740 | 0.894 | 0.166 | -1.325 |
| | (±0.001) | (±0.027) | (±0.102) | (±4.041) | (±0.065) | (±0.089) | (±2.444) |
| Fertility | | | | | | | |

Table 10 continued from previous page

| Dataset | Training Data (80%) | | | | Test Data (20%) | | |
|---|---|---|---|---|---|---|---|
| | Runtime | Precision | Coverage | VI | Precision | Coverage | VI |
| - OPDT | 0.25 | 1.000 | 0.429 | 34.300 | 0.971 | 0.480 | 6.600 |
| | (±0.132) | (±0.000) | (±0.024) | (±1.947) | (±0.049) | (±0.098) | (±4.477) |
| - BSCCART | 0.02 | 1.000 | 0.429 | 34.300 | 0.971 | 0.480 | 6.600 |
| | (±0.003) | (±0.000) | (±0.024) | (±1.947) | (±0.049) | (±0.098) | (±4.477) |
| - RSCRULES | 0.01 | 0.969 | 0.342 | 17.400 | 0.912 | 0.405 | 0.100 |
| | (±0.001) | (±0.024) | (±0.095) | (±3.950) | (±0.072) | (±0.119) | (±5.021) |
| - BRS | 0.27 | 0.997 | 0.259 | 18.713 | 0.910 | 0.293 | 1.025 |
| | (±0.025) | (±0.006) | (±0.047) | (±4.000) | (±0.075) | (±0.074) | (±3.001) |
| - IDS | 0.19 | 1.000 | 0.254 | 20.300 | 0.965 | 0.255 | 3.100 |
| | (±0.003) | (±0.000) | (±0.034) | (±2.710) | (±0.082) | (±0.096) | (±3.604) |
| - IREP | 0.03 | 0.958 | 0.375 | 16.192 | 0.954 | 0.371 | 3.577 |
| | (±0.001) | (±0.019) | (±0.053) | (±4.380) | (±0.024) | (±0.046) | (±1.860) |
| - PRISM | 0.13 | 1.000 | 0.188 | 15.000 | 0.976 | 0.290 | 3.800 |
| | (±0.029) | (±0.000) | (±0.037) | (±2.981) | (±0.052) | (±0.115) | (±3.327) |
| - Ripper | 0.04 | 0.969 | 0.333 | 17.182 | 0.876 | 0.206 | -1.939 |
| | (±0.001) | (±0.009) | (±0.044) | (±3.462) | (±0.052) | (±0.100) | (±2.228) |
| Heart Disease | | | | | | | |
| - OPDT | 524.81 | 0.979 | 0.165 | 29.200 | 0.800 | 0.180 | -9.200 |
| | (±534.168) | (±0.021) | (±0.047) | (±3.155) | (±0.063) | (±0.079) | (±5.160) |
| - BSCCART | 0.14 | 0.926 | 0.269 | 16.800 | 0.944 | 0.250 | 5.000 |
| | (±0.019) | (±0.011) | (±0.021) | (±6.339) | (±0.074) | (±0.058) | (±13.199) |
| - RSCRULES | 0.03 | 0.969 | 0.137 | 20.500 | 0.929 | 0.118 | -0.900 |
| | (±0.005) | (±0.020) | (±0.052) | (±3.171) | (±0.097) | (±0.064) | (±7.937) |
| - BRS | 0.36 | 0.949 | 0.210 | 15.010 | 0.928 | 0.200 | 0.636 |
| | (±0.049) | (±0.037) | (±0.079) | (±17.335) | (±0.089) | (±0.081) | (±11.493) |
| - IDS | 0.20 | 1.000 | 0.064 | 15.100 | 0.955 | 0.068 | 2.100 |
| | (±0.002) | (±0.000) | (±0.015) | (±3.510) | (±0.096) | (±0.020) | (±4.175) |
| - IREP | 0.06 | 0.903 | 0.224 | -5.888 | 0.827 | 0.342 | -19.485 |
| | (±0.001) | (±0.020) | (±0.039) | (±11.122) | (±0.043) | (±0.025) | (±10.521) |
| - PRISM | 1.01 | 1.000 | 0.056 | 13.300 | 0.922 | 0.053 | 0.200 |
| | (±0.040) | (±0.000) | (±0.008) | (±1.889) | (±0.130) | (±0.022) | (±4.442) |
| - Ripper | 0.10 | 0.944 | 0.163 | 13.592 | 0.852 | 0.208 | -4.524 |
| | (±0.001) | (±0.010) | (±0.046) | (±5.094) | (±0.042) | (±0.037) | (±5.392) |
| Heart Failure | | | | | | | |
| - OPDT | 729.32 | 0.982 | 0.331 | 62.100 | 0.935 | 0.335 | 5.100 |
| | (±581.965) | (±0.017) | (±0.088) | (±5.859) | (±0.045) | (±0.124) | (±9.036) |
| - BSCCART | 0.36 | 0.927 | 0.513 | 29.600 | 0.884 | 0.530 | -1.200 |
| | (±0.008) | (±0.012) | (±0.178) | (±11.937) | (±0.053) | (±0.178) | (±12.136) |
| - RSCRULES | 0.03 | 0.975 | 0.097 | 16.300 | 0.817 | 0.097 | -5.200 |
| | (±0.004) | (±0.023) | (±0.028) | (±4.296) | (±0.114) | (±0.064) | (±4.211) |
| - BRS | 0.30 | 0.953 | 0.238 | -41.974 | 0.782 | 0.239 | -19.254 |
| | (±0.062) | (±0.096) | (±0.318) | (±126.072) | (±0.106) | (±0.333) | (±30.173) |
| - IDS | 0.19 | 1.000 | 0.081 | 19.400 | 0.975 | 0.080 | 3.800 |
| | (±0.003) | (±0.000) | (±0.008) | (±2.011) | (±0.079) | (±0.029) | (±3.853) |
| - IREP | 0.07 | 0.975 | 0.103 | 17.578 | 0.823 | 0.123 | -8.545 |
| | (±0.002) | (±0.018) | (±0.012) | (±5.046) | (±0.064) | (±0.037) | (±8.333) |
| - PRISM | 0.82 | 1.000 | 0.087 | 20.700 | 0.955 | 0.067 | 1.000 |
| | (±0.078) | (±0.000) | (±0.012) | (±2.908) | (±0.096) | (±0.039) | (±4.944) |
| - Ripper | 0.11 | 0.994 | 0.096 | 21.412 | 0.792 | 0.088 | -6.730 |
| | (±0.002) | (±0.012) | (±0.006) | (±3.611) | (±0.063) | (±0.025) | (±3.177) |
| Hepatitis | | | | | | | |
| - OPDT | 0.82 | 0.998 | 0.675 | 42.200 | 0.907 | 0.656 | 0.500 |
| | (±0.727) | (±0.006) | (±0.059) | (±3.393) | (±0.082) | (±0.107) | (±7.792) |
| - BSCCART | 0.10 | 0.986 | 0.736 | 40.100 | 0.885 | 0.737 | -2.200 |
| | (±0.028) | (±0.013) | (±0.053) | (±3.755) | (±0.069) | (±0.121) | (±8.804) |
| - RSCRULES | 0.01 | 0.947 | 0.664 | 17.500 | 0.873 | 0.662 | -5.400 |
| | (±0.002) | (±0.024) | (±0.198) | (±4.223) | (±0.091) | (±0.238) | (±8.449) |
| - BRS | 0.27 | 1.000 | 0.458 | 29.314 | 0.962 | 0.446 | 3.429 |
| | (±0.022) | (±0.000) | (±0.187) | (±11.940) | (±0.049) | (±0.186) | (±3.645) |
| - IDS | 0.19 | 1.000 | 0.391 | 25.000 | 0.915 | 0.381 | 1.100 |
| | (±0.027) | (±0.000) | (±0.085) | (±5.416) | (±0.133) | (±0.133) | (±7.310) |
| - IREP | 0.06 | 0.967 | 0.646 | 26.330 | 0.970 | 0.475 | 4.554 |
| | (±0.001) | (±0.014) | (±0.054) | (±4.966) | (±0.020) | (±0.143) | (±2.280) |
| - PRISM | 0.15 | 1.000 | 0.341 | 21.800 | 0.881 | 0.294 | 2.700 |
| | (±0.017) | (±0.000) | (±0.133) | (±8.496) | (±0.312) | (±0.213) | (±1.889) |

Table 10 continued from previous page

| Dataset | Training Data (80%) | | | | Test Data (20%) | | |
|---|---|---|---|---|---|---|---|
| | Runtime | Precision | Coverage | VI | Precision | Coverage | VI |
| - Ripper | 0.07 | 0.973 | 0.615 | 28.004 | 0.988 | 0.430 | 5.664 |
| | (±0.001) | (±0.012) | (±0.045) | (±4.501) | (±0.024) | (±0.174) | (±2.723) |
| Indian Liver Patient | | | | | | | |
| - OPDT | 999.70 | 0.981 | 0.221 | 82.500 | 0.948 | 0.239 | 12.700 |
| | (±224.994) | (±0.008) | (±0.024) | (±10.069) | (±0.043) | (±0.028) | (±10.843) |
| - BSCCART | 0.61 | 0.942 | 0.297 | 54.500 | 0.907 | 0.313 | 2.300 |
| | (±0.015) | (±0.022) | (±0.040) | (±24.968) | (±0.036) | (±0.056) | (±13.679) |
| - RSCRULES | 0.04 | 0.996 | 0.044 | 19.600 | 0.986 | 0.045 | 4.200 |
| | (±0.015) | (±0.013) | (±0.006) | (±2.716) | (±0.045) | (±0.013) | (±2.860) |
| - BRS | 0.27 | 0.731 | 0.902 | -790.932 | 0.730 | 0.896 | -198.454 |
| | (±0.043) | (±0.031) | (±0.088) | (±96.273) | (±0.032) | (±0.087) | (±24.312) |
| - IDS | 0.21 | 1.000 | 0.038 | 17.400 | 0.975 | 0.039 | 3.500 |
| | (±0.003) | (±0.000) | (±0.005) | (±2.366) | (±0.079) | (±0.017) | (±3.866) |
| - IREP | 0.09 | 0.995 | 0.104 | 42.624 | 0.959 | 0.113 | 7.045 |
| | (±0.002) | (±0.005) | (±0.008) | (±7.690) | (±0.021) | (±0.021) | (±5.418) |
| - PRISM | 2.98 | 1.000 | 0.039 | 18.100 | 1.000 | 0.038 | 4.400 |
| | (±0.054) | (±0.000) | (±0.006) | (±2.923) | (±0.000) | (±0.022) | (±2.591) |
| - Ripper | 0.14 | 0.999 | 0.101 | 46.088 | 0.775 | 0.064 | -7.918 |
| | (±0.001) | (±0.001) | (±0.001) | (±0.820) | (±0.060) | (±0.014) | (±3.601) |
| Parkinsons | | | | | | | |
| - OPDT | 863.11 | 0.999 | 0.532 | 82.000 | 0.919 | 0.508 | 2.800 |
| | (±340.952) | (±0.003) | (±0.032) | (±4.447) | (±0.052) | (±0.077) | (±9.682) |
| - BSCCART | 1.77 | 0.977 | 0.597 | 70.200 | 0.884 | 0.621 | -5.800 |
| | (±0.116) | (±0.018) | (±0.077) | (±10.633) | (±0.070) | (±0.109) | (±18.207) |
| - RSCRULES | 0.03 | 0.976 | 0.240 | 28.500 | 0.990 | 0.246 | 8.600 |
| | (±0.007) | (±0.009) | (±0.014) | (±3.689) | (±0.032) | (±0.058) | (±3.777) |
| - BRS | 0.37 | 1.000 | 0.335 | 52.247 | 0.978 | 0.315 | 9.064 |
| | (±0.061) | (±0.000) | (±0.058) | (±9.034) | (±0.027) | (±0.073) | (±4.337) |
| - IDS | 0.19 | 1.000 | 0.163 | 25.400 | 0.876 | 0.154 | 0.000 |
| | (±0.004) | (±0.000) | (±0.033) | (±5.103) | (±0.166) | (±0.091) | (±4.397) |
| - IREP | 0.35 | 0.998 | 0.116 | 17.560 | 0.983 | 0.120 | 4.056 |
| | (±0.002) | (±0.003) | (±0.009) | (±1.554) | (±0.012) | (±0.018) | (±0.831) |
| - PRISM | 1.56 | 1.000 | 0.171 | 26.700 | 0.954 | 0.185 | 3.200 |
| | (±0.140) | (±0.000) | (±0.030) | (±4.668) | (±0.062) | (±0.067) | (±3.910) |
| - Ripper | 0.41 | 1.000 | 0.117 | 18.164 | 1.000 | 0.116 | 4.599 |
| | (±0.003) | (±0.000) | (±0.010) | (±1.515) | (±0.001) | (±0.018) | (±0.687) |
| Pima Indians Diabetes | | | | | | | |
| - OPDT | 1047.67 | 0.979 | 0.187 | 89.100 | 0.925 | 0.201 | 4.900 |
| | (±76.896) | (±0.012) | (±0.032) | (±6.590) | (±0.049) | (±0.047) | (±12.758) |
| - BSCCART | 0.76 | 0.922 | 0.297 | 36.200 | 0.889 | 0.309 | -8.400 |
| | (±0.010) | (±0.039) | (±0.105) | (±46.492) | (±0.038) | (±0.117) | (±21.287) |
| - RSCRULES | 0.07 | 0.962 | 0.105 | 37.400 | 0.953 | 0.097 | 4.900 |
| | (±0.003) | (±0.016) | (±0.041) | (±8.897) | (±0.047) | (±0.055) | (±7.340) |
| - BRS | 0.35 | 0.727 | 0.859 | -920.728 | 0.722 | 0.863 | -237.896 |
| | (±0.025) | (±0.006) | (±0.022) | (±36.483) | (±0.022) | (±0.037) | (±36.960) |
| - IDS | 0.22 | 1.000 | 0.035 | 21.500 | 0.963 | 0.037 | 3.700 |
| | (±0.008) | (±0.000) | (±0.010) | (±6.042) | (±0.078) | (±0.019) | (±5.250) |
| - IREP | 0.08 | 0.938 | 0.100 | 22.686 | 0.837 | 0.093 | -9.601 |
| | (±0.001) | (±0.013) | (±0.005) | (±6.723) | (±0.026) | (±0.016) | (±3.598) |
| - PRISM | 3.35 | 1.000 | 0.037 | 22.800 | 0.924 | 0.049 | 1.600 |
| | (±0.137) | (±0.000) | (±0.015) | (±8.942) | (±0.107) | (±0.021) | (±8.859) |
| - Ripper | 0.13 | 0.980 | 0.064 | 26.876 | 0.802 | 0.056 | -8.958 |
| | (±0.003) | (±0.015) | (±0.019) | (±5.161) | (±0.057) | (±0.020) | (±4.647) |
| Thoracic Surgery | | | | | | | |
| - OPDT | 133.21 | 0.949 | 0.264 | 46.200 | 0.855 | 0.247 | -9.800 |
| | (±327.588) | (±0.012) | (±0.058) | (±5.095) | (±0.079) | (±0.037) | (±16.430) |
| - BSCCART | 0.10 | 0.991 | 0.064 | 20.200 | 0.521 | 0.066 | -3.800 |
| | (±0.047) | (±0.015) | (±0.052) | (±15.483) | (±0.449) | (±0.054) | (±4.392) |
| - RSCRULES | 0.02 | 0.937 | 0.244 | 31.700 | 0.867 | 0.214 | -4.900 |
| | (±0.006) | (±0.010) | (±0.084) | (±7.009) | (±0.059) | (±0.091) | (±13.304) |
| - BRS | 2.13 | 0.898 | 0.706 | -21.068 | 0.863 | 0.717 | -27.672 |
| | (±5.802) | (±0.015) | (±0.127) | (±45.358) | (±0.023) | (±0.143) | (±12.292) |
| - IDS | 0.21 | 1.000 | 0.038 | 14.400 | 0.933 | 0.031 | -0.100 |
| | (±0.003) | (±0.000) | (±0.011) | (±4.169) | (±0.200) | (±0.022) | (±8.950) |

Table 10 continued from previous page

| Dataset | Training Data (80%) | | | | Test Data (20%) | | |
|---|---|---|---|---|---|---|---|
| | Runtime | Precision | Coverage | VI | Precision | Coverage | VI |
| - IREP | 0.05 | 0.902 | 0.322 | -2.136 | 0.862 | 0.445 | -12.724 |
| | ($\pm$0.002) | ($\pm$0.014) | ($\pm$0.049) | ($\pm$18.640) | ($\pm$0.016) | ($\pm$0.072) | ($\pm$4.690) |
| - PRISM | 1.70 | 1.000 | 0.045 | 16.900 | 1.000 | 0.048 | 4.500 |
| | ($\pm$0.090) | ($\pm$0.000) | ($\pm$0.005) | ($\pm$1.912) | ($\pm$0.000) | ($\pm$0.020) | ($\pm$1.841) |
| - Ripper | 2.16 | 0.914 | 0.250 | 26.356 | 0.835 | 0.130 | -8.167 |
| | ($\pm$6.556) | ($\pm$0.023) | ($\pm$0.044) | ($\pm$10.267) | ($\pm$0.069) | ($\pm$0.050) | ($\pm$5.930) |

# B   Feature Grouping Using ML-Based Feature Importance

As a concrete demonstration that prior information extracted from ML can help feature grouping, we select the German dataset, which is the largest dataset among those in Table 3. Suppose that we need to find a pattern complying with the `{categorical}-{numerical}` structure. First, feature importance is extracted from a LightGBM (LGBM) model (Ke et al., 2017) trained with 1,000 boosting iterations. Figure 4 presents the feature importance scores evaluated by two complementary metrics: information gain and split count. Based on this prior information, we select the top six features from each metric, as highlighted in red in Figure 4, to define two feature groups. Table 11 shows the VI improvement over time with a runtime limit of 600 seconds. As shown in Table 11, OPDT with LGBM reaches the optimal VI of 41.0 in 5 seconds, whereas OPDT Only reaches the same optimal VI in 23 seconds, and reduces the total runtime to prove optimality from 446.4 seconds to 90.8 seconds, which is a fivefold reduction in runtime. This experiment demonstrates a heuristic approach in which an ML method is used to extract feature importance rankings to define feature groups for the BSC framework.

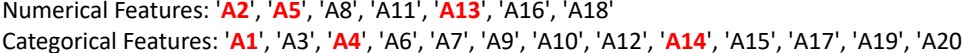

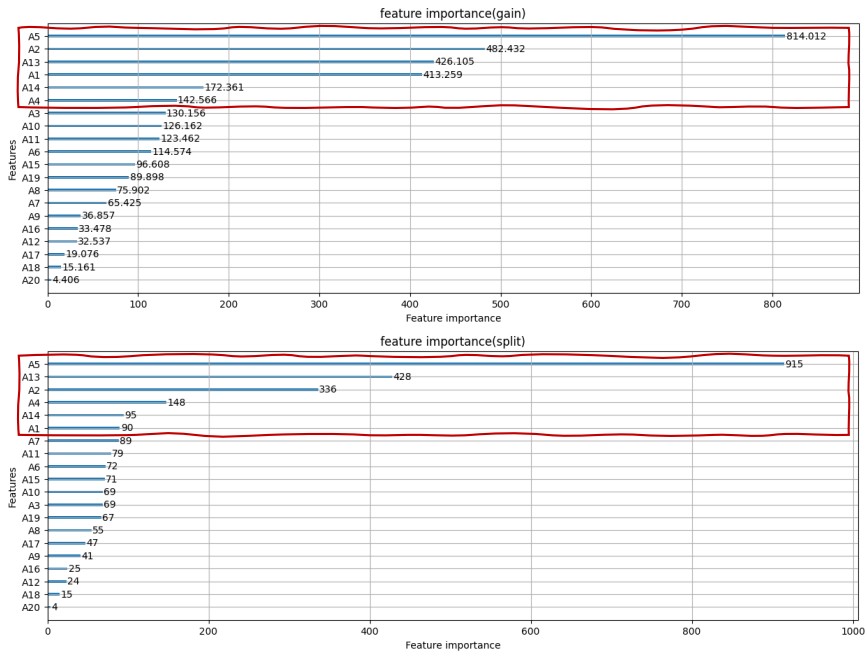

Figure 4: Feature grouping using LGBM feature importance (gain and split) on the German dataset. Features are ranked by information gain (top) and split count (bottom). The numerical and categorical features within the top six (highlighted in red) are selected for the `{cat}-{num}` rule.

Table 11: VI improvement over time on the German dataset for the `{cat}-{num}` rule. OPDT with LGBM denotes OPDT using feature groups selected by LightGBM feature importance, while OPDT Only denotes OPDT without feature grouping. Each row shows the step at which VI improves and the corresponding elapsed time in seconds.

| Step | OPDT with LGBM | | OPDT Only | |
|------|------|------------|------|------------|
| | VI | Time (sec) | VI | Time (sec) |
| 1 | 0.0 | 0 | 0.0 | 0 |
| 2 | 1.0 | 1 | 2.0 | 1 |
| 3 | 2.0 | 1 | 5.0 | 2 |
| 4 | 4.0 | 2 | 14.0 | 5 |
| 5 | 5.0 | 2 | 20.0 | 21 |
| 6 | 12.0 | 3 | 41.0 | 23 |
| 7 | 41.0 | 5 | — | — |
| **Optimal** | **41.0** | **90.8** | **41.0** | **446.4** |

