# OpenReview forum: "Optimal Pattern Detection Tree for Symbolic Rule-Based Classification"
_TMLR — Accepted by TMLR_

### Review · Reviewer_83EL · 2026-02-03

**Summary Of Contributions:**

This manuscript introduces a rule-based method for binary classification termed the Optimal Pattern Detection Tree (OPDT). OPDT is based on a *structure-constrained* approach, where the features form multiple feature groups, and each node in the decision tree can only query features in a specific feature group. The goal is to find a rule that maximizes its Volume-Impurity (VI) index, which is the number of training examples covered by the rule minus a parameter $w$ times the number of misclassified examples. The authors formulated a Mixed-Integer Program (MIP) for this problem, and the method is empirically shown to have comparable or better performance compared to baselines on several datasets.

**Audience:**

No

**Audience Explanation:**

While I found it interesting to use domain knowledge to guide the structure and feature selection in rule-based ML, I would doubt whether the present work would be of great interest given the (arguably) straightforward method and the resulting computational overhead. Given the problem definition, the OPDT method seems to be a natural MIP formulation of the VI index objective. The method takes significantly more time than the baselines (Table 6) on arguably small datasets (Table 3). This is partly because the MIP contains at least as many variables as the number of nodes ($2^D$) times the number of training examples.

I have a question related to the comment above: Since the MIP in (1)-(20) maximizes the highest VI index among all leaves of the tree, why do we need to solve for the entire decision tree? (Intuitively, all nodes that are not ancestors of the leaf with the largest VI index are "inactive" in the solution.) An alternative would be to enumarate all the leaf nodes of the tree and, for each leaf node $\ell$, solve a separate MIP to maximize the VI index at $\ell$. This should reduce the number of nodes in each MIP from $2^D$ to $D$.

**Broader Impact Concerns:**

None.

**Claims And Evidence:**

No

**Claims Explanation:**

The contribution of the paper could be presented more clearly if the authors put this work into the context of prior related work better. Below are some concrete questions:

- **Volume-Impurity Index:** How did prior work evaluate the "effectiveness" of a decision rule? Are those metrics identical or related to the VI index? If not, what is the benefit of using this new metric?

- **The MIP:** Is the MIP in Equations (1)-(20) largely similar or identical to previous MIP-based methods for decision rules/lists/trees? If not, what is "new" in the current formulation (apart from the VI objective and the "Branching Structure Constraints" added in Algorithm 1)?

In addition, as detailed in the "Requested Changes" section, the current manuscript seems to contain numerous inaccuracies in the presentation of the OPDT method. I would urge the authors to carefully proofread the maths before re-submitting the manuscript (or submitting a revision of it).

**Requested Changes:**

Writing suggestions:

- The notion of a "rule" (considered in this paper) should be formally defined somewhere, e.g., as a root-to-leaf path in a decision tree, or an AND of the conditions induced by all non-leaf nodes on that path.

- When introducing the VI index, it should be mentioned that the goal is to find a rule that maximizes this index. (The current writing might suggest that the VI index is used as a proxy for another objective.)

- Section 3.1: Please clarify what it means (mathematically) for features to "have mutually exclusive impacts on classification". (The meaning wasn't perfectly clear even after reading the example in the next sentence.)

- Figure 1: The figure is ambiguous in various aspects: (1) In the left part, why is $G_1$ surrounded by "{}"? (2) Can different feature groups be overlapping, as the left part suggests (e.g., $G_1$ and $G_4$)? (3) In the middle part, what are the numbers $0$ through $15$?

- Bottom of Page 5: "$G_1, G_2, \cdots, G_D$" $\to$ "$G_1, G_2, \ldots, G_D$" ("\cdots" vs "\ldots").

- Figure 2: This figure was particular confusing---it differs from Figure 1 and the MIP formulation in that it enforces the same feature group at each level of the decision tree, while Figure 1 and the MIP allow each node to have a different feature group. Is this special case ever considered in the rest of the paper?

- Table 1: $N$ and $K$ are used both as sets and as numbers.

- Table 1: Is $G_A$ the same as $P$? If so, why is a different notation needed?

- Table 1: "Numerical feature group" and "categorical feature group" are not introduced.

- Table 1: "$i \in 1, \ldots, N$" should be $i \in N$, if $N$ is a set.

- Table 1: $Y_{ik}$ should be introduced after $K$ and $k$.

- Table 1 (continued): If $T_B$ is a set, there should be "{}" around "$1, \ldots, \lfloor T/2 \rfloor$". Same comment for $T_L$.

- Table 1 (continued): What is the "objective function" when defining $T_{\mathrm{obj}}$?

- The MIP in (1)-(20): The "type" of each variable---in addition to those in (19) and (20)---should be specified, i.e., whether each of them is $\{0, 1\}$-, negative-, or real-valued.

- Equation (9): What are $\vec{\epsilon}$ and $\epsilon_{\mathrm{max}}$?

- Equations (13) and (14): What is $M$?

- Equations (16)-(18): These constraints seem to be enforcing that $I_{\mathrm{max}}$ is the maximum VI index over all leaves. However, I don't think they would function as intended. For instance, we could have $q_t = 1$ for the leaf $t$ with the maximum VI index, and $q_t = 0$ elsewhere. Then, $I_{\mathrm{max}}$ could be as high as the maximum VI index plus $M$. (A simple workaround would be setting $I_{\mathrm{max}}$ as a weighted sum of the VI indices by $q$ and enforcing that $q \ge 0$ entrywise.)

- Algorithm 1: It should be stated more clearly that Lines 6, 8, and 10 are constraints to be added to the MIP, rather than "usual" statements in a pseudocode.

- Algorithm 1: It would be clearer if you write "$a_{pt} = 0$" instead of "$a_{pt} \le 0$" on Lines 8 and 10.

---

> ### Author Response · Authors · 2026-02-17
> **Response (part 01) to Reviewer Comments**
>
> Dear Reviewer,
>
> Thank you for your meaningful comments and thorough review of our paper.
>
> We first respond to your high-level critical comments below. We are currently revising the paper to update details for figures, tables, equations, and notations based on your Requested Changes. We will submit the updated manuscript once we complete the revision.
>
> ***Reviewer Comment*** - Volume-Impurity Index: How did prior work evaluate the "effectiveness" of a decision rule? Are those metrics identical or related to the VI index? If not, what is the benefit of using this new metric?
>
> **Response** We appreciate your valuable feedback on this. In response, we have revised the Metrics paragraph in Section 3.2 to clarify this more thoroughly. The following paragraph will be added.
>
> *Prior RBML methods employ heuristic approaches with no optimality guarantees, focusing on precision maximization while treating coverage as a byproduct (Cendrowska, 1988), balancing precision with rule complexity such as the number of rules and conditions per rule (Lakkaraju et al., 2016), adopting a two-phase strategy that prioritizes coverage in the growing phase before shifting to precision improvement through pruning on separate validation sets (Fürnkranz & Widmer, 1994; Cohen et al., 1995), or balancing accuracy and interpretability through Bayesian likelihood and prior terms (Wang et al., 2017).*
>
> *Unlike these traditional implicit or sequential evaluation approaches employed in heuristic rule learning, we define a novel metric specifically designed for optimization-based frameworks that explicitly balances coverage volume with misclassification control through the weight parameter $w$, allowing domain experts to specify misclassification tolerance directly within the objective function before model training.*
>
> ***Reviewer Comment*** - The MIP: Is the MIP in Equations (1)-(20) largely similar or identical to previous MIP-based methods for decision rules/lists/trees? If not, what is "new" in the current formulation (apart from the VI objective and the "Branching Structure Constraints" added in Algorithm 1)?
>
> **Response** Our formulation adopts three fundamental constraints from Bertsimas and Dunn (2017)'s OCT: tree structure constraints (split decisions at branching nodes), sample routing constraints (directing samples from root to leaf), and misclassification constraints. However, we introduce several critical innovations: (1) Constraints (4-5) force samples to rightmost leaf nodes when parent nodes do not split, correcting sample routing behavior for accurate VI evaluation at branching-terminated parent nodes; (2) Constraints (16-18) linearize the inherently nonlinear maximal VI selection among leaf nodes, representing a non-trivial technical contribution; (3) we introduce the BSCCART model to provide warm-start solutions for the BSC framework since traditional decision tree methods produce infeasible solutions for our OPDT formulation; (4) Section 4.2 demonstrates computational tractability through three enhancement techniques (including BSCCART as warm-start initialization) despite the problem's NP-hard nature. These contributions collectively enable the first provably optimal pattern discovery method from data with domain knowledge integration capabilities.

---

> ### Author Response · Authors · 2026-02-17
> **Response (part 02) to Reviewer Comments**
>
> ***Reviewer Comment*** - While I found it interesting to use domain knowledge to guide the structure and feature selection in rule-based ML, I would doubt whether the present work would be of great interest given the (arguably) straightforward method and the resulting computational overhead. Given the problem definition, the OPDT method seems to be a natural MIP formulation of the VI index objective. The method takes significantly more time than the baselines (Table 6) on arguably small datasets (Table 3). This is partly because the MIP contains at least as many variables as the number of nodes ($2^D$) times the number of training examples.
>
> **Response** We sincerely appreciate your thoughtful feedback regarding computational overhead and method novelty. We would like to respectfully address these important concerns.
>
> ### Regarding Computational Overhead:
> - We appreciate your observation about runtime comparisons. We would like to respectfully note that OPDT's reported runtime includes the time required to prove optimality—a guarantee that heuristic methods do not provide. When optimality requirements are relaxed slightly, Table 9 demonstrates that OPDT typically identifies high-quality solutions within the first 2 minutes (As noted on page 13). Also, we provide convincing computational results showing the quality of solutions is superior to those from other methods on both multiple shuffled training and test datasets.
>
> - While we acknowledge that heuristic methods produce solutions more quickly, they provide no mechanism for solution improvement or quality guarantees after the solution returns. We believe that there are high-stakes domains such as finance, healthcare, and law where accountability and reliability of a ML solution are critical. To our knowledge, this is the first work demonstrating empirically that provably optimal pattern discovery can be achieved within 10 minutes (rather than hours or days) for moderate-sized datasets.
>
> ### Regarding Method Novelty and Practical Value:
>  - We fully acknowledge the inherent computational limitations of optimization-based models, as thoroughly discussed in Bertsimas and Dunn (2017). However, we respectfully suggest that our contribution extends beyond the MIP formulation itself. The BSC framework represents a paradigm shift that enables domain knowledge integration and can complement any tree-based heuristic method.
> - For instance, if computational time is a primary concern, decision makers can first apply any heuristic RBML method (e.g., we develop BSCCART as an example for this complementary concept since no prior method exists for our structure-constrained paradigm) to obtain a tree-structured solution, then use that solution to define structural constraints for OPDT via the BSC framework. OPDT can then either improve upon the heuristic solution or provide evidence that it is near-optimal given the structural constraints. This hybrid approach combines the computational efficiency of heuristic methods with the optimality guarantees of optimization-based approaches. We implicitly give a hint about the potential value of our method as a complementary tool for other ML methods in Section 3.
> - Given this hybrid paradigm, this paper provides computational evidence that investing an additional 10 minutes after obtaining a solution from any RBML method yields substantial benefits: (1) OPDT can find provably optimal solutions for datasets with runtime under 600 seconds (Table 6), (2) even without proven optimality, OPDT identifies superior solutions within approximately 2 minutes, and (3) when improvement is not possible, OPDT provides quality verification for the heuristic solution through optimality gap.
> - We believe this methodology offers a meaningful enhancement to the quality and reliability of ML solutions in ways that may be difficult to achieve without an optimization framework. The value lies not merely in the MIP formulation itself, but in providing the first provably optimal pattern discovery method with domain knowledge integration capabilities for high-stakes decision-making contexts where solution quality and accountability are essential.

---

> ### Author Response · Authors · 2026-02-17
> **Response (part 03) to Reviewer Comments**
>
> ***Reviewer Comment*** - I have a question related to the comment above: Since the MIP in (1)-(20) maximizes the highest VI index among all leaves of the tree, why do we need to solve for the entire decision tree? (Intuitively, all nodes that are not ancestors of the leaf with the largest VI index are "inactive" in the solution.) An alternative would be to enumarate all the leaf nodes of the tree and, for each leaf node l, solve a separate MIP to maximize the VI index at l. This should reduce the number of nodes in each MIP from $2^D$ to $D$.
>
> **Response** We thank you for this thoughtful suggestion. Upon careful consideration, we note that separating a large MIP into smaller subproblems does not always guarantee computational improvement. While your proposed approach reduces the size of each individual MIP from $2^D$ nodes to $D$ nodes, it creates exponentially many subproblems (e.g., $2^D$) in return, eliminating any potential computational gain.
>
> More generally, this decomposition approach does not work for combinatorial optimization problems. As a simple counterexample, consider an NP-hard sequential problem like OPDT. If the decomposition approach were effective, one could argue: "instead of solving the NP-hard problem in a single MIP formulation, split it into small deterministic cost functions for each sequence and solve each sequence problem with a for-loop." However, this approach is obviously slower. The fundamental reasons are:
>
> - Constraint Propagation: MIP solvers exploit structural dependencies to infer variable values across the entire tree. This global information would be lost in separate formulations, preventing the solver from leveraging relationships between different parts of the solution space.
>
> - Shared Bounds: Modern solvers use branch-and-bound techniques that benefit from global optimization across all nodes simultaneously, enabling more aggressive pruning than isolated subproblems. The solver can eliminate large portions of the search space by reasoning about the entire tree structure, which is impossible when solving independent subproblems.
>
> Therefore, decomposing a problem into exponentially many subproblems does not guarantee computational improvement and typically degrades performance.
>
> However, your question raises a meaningful point that highlights why the BSC framework is a good approach: it provides a unified mathematical formulation that implicitly covers your proposed approach as well. For example, suppose we construct a depth-2 tree to explore all negation combinations (i.e., $\geq$ or $<$ for each split). We can create four separate BSC configurations: $B(split) = [1, 1, 0]$ with $L(target) = [1, 0, 0, 0]$; $B(split) = [1, 1, 0]$ with $L(target) = [0, 1, 0, 0]$; $B(split) = [1, 0, 1]$ with $L(target) = [0, 0, 1, 0]$; and $B(split) = [1, 0, 1]$ with $L(target) = [0, 0, 0, 1]$. This decomposed configuration is equivalent to your proposed enumeration approach.
> The reason we maintain the MIP formulation for the full binary tree is that it provides greater flexibility to represent various cases, including your proposed approach, without tailoring the mathematical formulation for each case. We aim to provide one unified mathematical formulation that remains consistent across different scenarios. Instead of modifying the mathematical formulation itself, the BSC framework allows us to modify the tree structure as needed, including your proposed approach as a special case. However, for empirical validation, this paper only demonstrates computational time with full negation combinations to verify worst-case runtime instead of assuming each special case.
>
> We hope this clarifies our design rationale, and we remain open to further updates on the manuscript. We will submit the updated manuscript within two weeks.

---

> ### Author Response · Authors · 2026-02-21
> **Response (part 04) to Reviewer Comments**
>
> Dear Reviewer,
>
> Thank you for your detailed and thoughtful feedback on our model. We sincerely appreciate the time and effort you invested in reviewing our work. Your thoughtful comments have been helpful in refining our manuscript.
>
>
> ***Reviewer Comment*** - The notion of a "rule" (considered in this paper) should be formally defined somewhere, e.g., as a root-to-leaf path in a decision tree, or an AND of the conditions induced by all non-leaf nodes on that path.
>
> **Response** We have added formal definitions of "rule" in the Approach subsection of Section 3.2 to address this point.
>
> * *We utilize a decision tree approach to discover this optimal rule, where each rule corresponds to a root-to-leaf path that maximizes the VI index through nested if-then-else structures handling negation scenarios while incorporating prior knowledge via structural constraints.*
>
> ***Reviewer Comment*** - When introducing the VI index, it should be mentioned that the goal is to find a rule that maximizes this index. (The current writing might suggest that the VI index is used as a proxy for another objective.)
>
> **Response** We have added the following sentence in the Metrics subsection of Section 3.2 to clarify this point. We expect that this addition explicitly establishes that maximizing the VI index is the primary objective of OPDT, not a proxy for another goal.
>
> * *The goal of OPDT is to identify a rule (i.e., root-to-leaf path in a decision tree) that maximizes the VI index in the leaf node.*
>
> ***Reviewer Comment*** - Section 3.1: Please clarify what it means (mathematically) for features to "have mutually exclusive impacts on classification". (The meaning wasn't perfectly clear even after reading the example in the next sentence.)
>
> **Response** We've revised it as follows. I hope it makes it clear than concern about potential ambiguity in the original.
>
> * *We define feature groups by splitting features into small subsets that capture a distinct semantic or conceptual aspect of the classification problem.*
>
> ***Reviewer Comment*** - Figure 1: The figure is ambiguous in various aspects
>
> **Response** We have added the following sentence in Section 3.1 (Structure Constrained Decision Tree) to clarify ambiguous statements. We have also updated the figure and its caption for clarity.
>
> * *As shown in Figure 1, our framework allows feature groups to be overlapping, providing flexibility in capturing features that may belong to multiple semantic or conceptual categories. Additionally, we define $G_{A}$ as a special feature group containing all features, which serves as a default group when prior domain knowledge is insufficient to establish meaningful feature groupings.*
>
> [Caption for Figure 01]
> * *Structure Constrained Decision Tree (SCDT). The tree uses a breadth-first indexing scheme where nodes are numbered sequentially from the root (node 0) through internal nodes to leaf nodes. Feature groups $(G_{A}, G_{1}, G_{2}, \dots)$ can be overlapping, where $G_{A}$ denotes a special feature group containing all features, used when prior domain knowledge for defining feature groups is unavailable.*
>
> ***Reviewer Comment*** - Bottom of Page 5: ("\cdots" vs "\ldots").
>
> **Response** Thank you for finding the typo. We have fixed it as you recommended: (e.g., $G_1, G_2, \dots ,G_D$)
>
>
> ***Reviewer Comment*** - Figure 2: This figure was particular confusing
>
> **Response**  We have added the following sentence in the Approach subsection of Section 3.2 and updated Figure 2 and its caption.
>
> * *While SCDT allows different feature groups at each branching node as shown in Figure 1, OPDT enforces the same feature group at each level of the decision tree to systematically explore all negation combinations (i.e., $\geq$ or $<$ for each feature within a feature group) across the chain of rule conditions.*
>
> [Caption for Figure 02]
> * *All nodes at the same depth are restricted to use identical feature groups for the chain of rule conditions $\\\{G_1 - G_2 - \dots - G_D\\\}$.*

---

> ### Author Response · Authors · 2026-02-21
> **Response (part 05) to Reviewer Comments**
>
> ***Reviewer Comment*** - Table 1
>
> **Response**  We have updated Tables 1 and 2, the MIP formulation, and the BSC algorithm to make them clear for readers.
>
> ***Reviewer Comment*** - The MIP equations
>
> **Response** We have updated it by adding the following sentences:
>
> * *Finally, decision variables (19), (20), and (21) specify the branching structure, where $d_t$ indicates whether node $t$ has a split, $a_{pt}$ selects features for the split, and $b_t$ defines the threshold for split $a^{T}x < b$ at each branch node. Variables (22), (23), and (24) determine sample assignments to leaf nodes, leaf node activation, and class predictions, respectively. Count variables (25), (26), and (27) track the number of samples in each leaf node, samples per class, and misclassified samples in target leaf nodes, respectively. Binary variable (28) indicates whether leaf node $t$ has the maximum VI index. Finally, $I_{\max}$ (29) represents the maximum VI index value across all target leaf nodes.*
>
> * We have added the following sentence next to Equation (9): "where $\vec{\epsilon}$ denotes a vector of sufficiently small positive values $\epsilon_p$ for each feature $p \in P$ and $\epsilon_{max} = \max_p \epsilon_p$."
>
> * We have added "*where $M$ denotes a big-M constant*" next to Equations (13) and (14) to explain that $M$ represents a sufficiently large number.
>
> ***Reviewer Comment*** - Algorithm 1
>
> **Response** We have added the following sentence to the Branching Structure Constraints subsection of Section 3.2.
>
> * *Finally, BSC adds additional constraints on $d_t$, $a_{pt}$, $b_t$, and $T_{obj}$ from Algorithm 1 to the main OPDT MIP formulation.*
> * We have updated $a_{pt}$ as you recommended: "$a_{pt} = 0$" instead of "$a_{pt} \leq 0$"
>
> ***Reviewer Comment*** - Equations (16)-(18): "I don't think they would function as intended. "
>
> **Response** We believe the equations work as intended. If we could have $q_t = 1$ for the leaf $t$ with the maximum VI index and $q_t = 0$ elsewhere, then $I_{\max}$ would equal the maximum VI index value without the penalty term $M$, since the $M$ term is multiplied by $(1 - q_t)$.

---

> > ### Comment · Reviewer_83EL · 2026-02-26
> > **A few follow-up questions**
> >
> > Thank you for the response! I have a few follow-up questions.
> >
> > **Splitting into $2^D$ MIPs.** The purpose of splitting is not to completely avoid the exponential dependence on $D$, but to (potentially) reduce the runtime. In particular, I would disagree with the claim that the existence of $2^D$ subproblems is "eliminating any potential computational gain".
> >
> > Suppose that the runtime for solving an $N$-variable MIP roughly scales as $N^{\alpha}$ for some exponent $\alpha > 1$. Then, the current approach---which involves $\approx 2^D$ variables---has a runtime that scales as $2^{\alpha D}$. In contrast, solving $2^D$ different MIPs (each with $\approx D$ variables) only gives a dependence of $2^D \cdot D^{\alpha} \ll 2^{\alpha D}$. The point is: having $2^D$ as the number of MIPs only leads to a linear-in-$2^D$ runtime, while having it as the number of variables leads to a super-linear blow-up (of $2^{\alpha D}$).
> >
> > Given that the splitting approach seems a natural and obvious improvement, it would cast doubt on the efficiency of the current tree-based approach, unless an empirical comparison between the two is given.
> >
> > **Feature groups.** The current wording "splitting features into small subsets ..." may suggest that the feature groups are disjoint. It is worth emphasizing that feature groups may overlap right after this sentence (instead of waiting until Figure 1). An alternative is to simply say something like "a feature group is a subset of all features".
> >
> > **The MIP.** I have two more questions on the MIP in (1)-(20). First, the meaning of parameters $\vec{\epsilon}$, $\epsilon_{\mathrm{max}}$, and $M$ are still a bit unclear. (They seem to be slack parameters that help enforce some constraints.) It should be clarified why they are necessary, and whether their values affect the objective of the MIP.
> >
> > Second, I still believe that Equations (16)-(18) do not function as intended. Here is a detailed example: Suppose that, for every $t \in T_{\mathrm{obj}}$, we have $N_t - wL_t = \alpha$, i.e., all leaves give the same VI index $\alpha$. Then, setting $q_t = 1 / |T_{\mathrm{obj}}|$ and $I_{\mathrm{max}} = \alpha + M \cdot (1 - 1 / |T_{\mathrm{obj}}|)$ would be feasible for (16)-(18). However, this leads to a value of $I_{\mathrm{max}}$ that is strictly larger than $\max_t(N_t - wL_t)$.
> >
> > Could you clarify whether this is indeed an error in the current MIP formulation and, if so, whether this impacts the empirical results reported in the manuscript?
> >
> > As mentioned in my initial review, a simple workaround would be replacing (16) and (17) with the equation
> >
> > $I_{\mathrm{max}} = \sum_{t \in T_{\mathrm{obj}}}q_t \cdot (N_t - wL_t),$
> >
> > which would encourage $q$ to concentrate on the entry with the highest VI index.

---

> > > ### Author Response · Authors · 2026-03-02
> > > **Response (part 01) to Reviewer Follow-up Questions**
> > >
> > > We deeply appreciate your detailed and insightful comments throughout the review process. The depth and rigor of your feedback represent a co-author-level contribution to this work. We sincerely appreciate the considerable effort you have invested in the review.
> > >
> > >
> > > ***Reviewer Comment*** - Splitting into $2^D$ MIPs.
> > >
> > > **Response** Thank you for the clear explanation about the reason of the splitting idea. We agree with your point, and we recognize that your splitting idea aligns well with the motivation behind feature grouping in this research.
> > >
> > > Our previous response was based on a theoretical perspective, and we simply note that it is not always guaranteed that decomposing a MIP into smaller subproblems reduces overall runtime. However, even if the above statement holds from a theoretical or complexity analysis perspective, we were also interested in whether the splitting approach yields practical computational gains like your splitting idea: does decomposing the problem into smaller variable subsets yield practical computational gains? We observe that the key difference between your proposed splitting idea and the experiment taken in this paper is that you focus on splitting by tree structure (leaf nodes), whereas this paper focuses on splitting features into feature groups for the computational experiments. Given the similarity between two splitting ideas, we believe this paper provides relevant (though not exactly equivalent) empirical evidence on your question. Rather than splitting by leaf nodes as you propose, we split features into numerical and categorical groups, creating four separate smaller cases (numerical-only, categorical-only, mixed, and all-features), and report the computational time for each in Table 4. We believe your splitting approach by leaf nodes would also show a similar result.
> > >
> > > For example, in the Chronic Kidney dataset, as you suggest, splitting a problem into four smaller cases shows significant computational gains in runtime compared to the all-features case. Moreover, not only this dataset, but the Echocardiogram and Fertility datasets also support a splitting idea. However, the Parkinson's and Heart Failure datasets do not follow this pattern, which may suggest that decomposition does not always reduce runtime. Nevertheless, we observe that Table 4 shows a general tendency toward reduced computation time, at least statistically, and this is precisely the point of this paper: to demonstrate practical computational gains and show that this optimization-based method is applicable to real-world-scale problems.
> > >
> > > We think your concern about the efficiency of the current tree-based approach implicitly assumes that OPDT always uses the $B^{(split)} = [1, 1, \ldots, 1]$ configuration. However, the BSC framework allows for covering various split scenarios. As explained in a previous response, if we use different $2^D$ BSC configurations for each smaller MIP (e.g., $B{(split)} = [1, 1, 0]$ with $L{(target)} = [1, 0, 0, 0]$ through $B{(split)} = [1, 0, 1]$ with $L{(target)} = [0, 0, 0, 1]$ for a depth-2 example), each problem can be solved under the OPDT formulation without the extra effort of formulating a different MIP model separately for each individual smaller MIP. We believe the tree-based MIP formulation for OPDT with BSC offers substantial value in terms of flexibility and adaptability, rather than being a cause of scalability issues.
> > >
> > > In this paper, we focused on feature grouping by data type (numerical and categorical) to observe practical computational time gains for a split idea under worst-case conditions (e.g., no prior knowledge about features, solving a full case search at once rather than iterative runs over several smaller problems). We want to show that even under these worst-case conditions, the time to solution is within 2 minutes, to counter the frequently raised criticism of the slow runtime of optimization-based tree models (e.g., prior OCT methods required hours, not minutes, for binary classification).
> > >
> > > We will carefully consider your proposed splitting idea on $B{(split)}$ as a promising direction for future work, particularly when extending this paper. We believe this would provide an additional compelling experimental case demonstrating computational gains (and mitigating scalability concerns) through the BSC framework, in addition to the current feature-grouping $B{(group)}$-based results presented in Table 4.

---

> > > ### Author Response · Authors · 2026-03-02
> > > **Response (part 02) to Reviewer Follow-up Questions**
> > >
> > > ***Reviewer Comment*** - Feature groups. The current wording "splitting features into small subsets ..." may suggest that the feature groups are disjoint. It is worth emphasizing that feature groups may overlap right after this sentence (instead of waiting until Figure 1). An alternative is to simply say something like "a feature group is a subset of all features".
> > >
> > >
> > > **Response** We appreciate this valuable suggestion. We recognize that the original wording "splitting" may mislead readers into assuming feature groups are disjoint. We have revised the definition in the manuscript as follows:
> > >
> > > *we define a feature group as a subset of features that captures a distinct semantic or conceptual aspect of the classification problem, where feature groups are not required to be disjoint and may overlap.*
> > >
> > >
> > >
> > > ***Reviewer Comment*** - The meaning of parameters $\epsilon$, $\epsilon_{max}$ and $M$ are still a bit unclear. (They seem to be slack parameters that help enforce some constraints.) It should be clarified why they are necessary, and whether their values affect the objective of the MIP.
> > >
> > >
> > > **Response** We have added the following description to the MIP formulation section to address this concern:
> > >
> > > *These $\epsilon_p$ and $\epsilon_{\max}$ arise from a technical necessity in MIP formulations: strict inequality constraints of the form $a_t^\top x_i < b_t$ are not directly supported by MIP solvers and must be converted to non-strict form. The parameter $\epsilon_{\max}$ acts as the big-M constant since the maximum possible value of $a_t^\top(x_i + \vec{\epsilon}) - b_t$ is $1 + \epsilon_{\max}$.*
> > >
> > > *Here, we take $M = |N|$ as a sufficiently large value since the misclassification loss at any leaf node is bounded by $|N|$.*
> > >
> > >
> > > ***Reviewer Comment*** - I still believe that Equations (16)-(18) do not function as intended.
> > >
> > >
> > > **Response** We appreciate this careful re-examination of Constraints (16)-(18). We respectfully note that the suggested simple workaround equation does not function as a direct replacement in the MIP formulation, as it is nonlinear. This paper derives an equivalent formulation to linearize it. Instead of using the simple equation, we needed to find an equivalent set of linear constraints, which are precisely Constraints (15)-(17) (the equation order is shifted in the revised version) and (27). Furthermore, given our computational outputs and empirical validation through comparisons with other algorithms, we are confident that these constraints constitute a correct formulation for OPDT. Otherwise, it would be difficult to obtain the results shown in Tables 6 and 8, and Appendix Table 9 empirically.
> > >
> > > Regarding your counterexample, we believe it happened due to our incomplete initially submitted version, in which we mentioned $q_t$ as a binary variable only in Table 2. This was because we originally targeted a shorter submission format and did not list all variable types explicitly in the MIP formulation, assuming Table 2 would convey the necessary information. Given your feedback, we have added Constraint (27) to make the binary nature of $q_t$ explicit for the reader. Then, given $q_t \in {0, 1}$, it is impossible to set $q_t$ to a fractional value. Only one $q_t$ should equal 1, and all others should equal 0. We again sincerely appreciate the exceptional level of detail in your feedback.

---

> > > > ### Comment · Reviewer_83EL · 2026-03-03
> > > >
> > > > Thank you for clarifying that the $q_t$s are binary variables! Equations (16)-(18) as well as the choice of $M = |N|$ make a lot of sense to me.
> > > >
> > > > Regarding $\epsilon$ and $\epsilon_{\mathrm{max}}$: Exactly how small should these parameters be? Is there an explicit condition for them (like $M = |N|$ for parameter $M$) under which the MIP objective exactly gives the maximum possible VI index?

---

> > > > > ### Author Response · Authors · 2026-03-04
> > > > > **Response to Reviewer Follow-up (2nd) Questions**
> > > > >
> > > > > ***Reviewer Comment*** Regarding $\epsilon_p$ and $\epsilon_{\max}$: Exactly how small should these parameters be? Is there an explicit condition for them under which the MIP objective exactly gives the maximum possible VI index?
> > > > >
> > > > >
> > > > > **Response** Thank you for your follow-up questions about the details we had still missed in describing the formulation clearly. We have added the following sentence in the revised version.
> > > > >
> > > > > *Practically, $\epsilon_p$ can be any sufficiently small positive number that does not cause numerical instabilities in the MIP solver. On the other hand, the largest valid value of $\epsilon_p$ is the smallest non-zero distance between adjacent values of feature $p$ in the data. We use the largest valid value of $\epsilon_p$ for numerical stability.*
> > > > >
> > > > > This is example (not included in the manuscript).
> > > > >
> > > > > For example, given numerical feature $p$ := $[0, 0.1, 0.4, 0.8, 1]$, the smallest non-zero difference is $0.1$ (the other differences are $0.3$, $0.4$, and $0.2$). Thus, any value of $\epsilon_p \leq 0.1$ is valid, provided it does not cause numerical instabilities in the MIP solver. If $\epsilon_p$ is set too small (e.g., $10^{-20}$), the solver may treat it as zero and fail to distinguish $\epsilon_p$ from $0$ (even though $\epsilon_p > 0$).
> > > > >
> > > > >
> > > > > Again, thank you for your high-quality review feedback on the formulation and for the impressive new splitting idea, given your careful reading of the manuscript.

---

> > > > > > ### Comment · Reviewer_83EL · 2026-03-04
> > > > > >
> > > > > > Thank you for the clarification and for revising the manuscript!

---

### Review · Reviewer_ZhSs · 2026-02-26

**Summary Of Contributions:**

The paper proposes Optimal Pattern Detection Trees (OPDT), a rule-based binary classification model that uses mixed integer programming (MIP) to discover an optimal symbolic pattern in data.
OPDT aims to find a rule that maximizes coverage of the target pattern while controlling misclassification.
A key contribution is the ability to enforce rule structures to meet user requirements, while still providing optimality guarantees from the underlying optimization formulation.
Experiments on a lot of datasets show that OPDT can produce high-quality, structure-compliant rules within reasonable runtime.

## Strengths:
1. Proposed model supports explicit structural constraints on rules, aligning outputs with prior knowledge and practical requirements.
2. Unlike heuristic rule learners, OPDT provides to optimal solutions under its formulation.
3. Experiments on 15 UCI datasets show that the proposed method is competitive across a wide range of healthcare and finance datasets.

## Weaknesses:
1. The MIP formulation is somewhat difficult to follow and validate, because it relies on a large number of inequality constraints and variables, which makes it hard for the reader to understand and verify the correctness of the encoding.
2. The paper includes only limited theoretical discussion of the MIP formulation, such as how the number of decision variables and constraints scales with the number of instances or features, which makes it difficult to assess the scalability of this MIP problem from a theoretical perspective.
3. Although the paper evaluates OPDT on many datasets, most of the benchmarks appear to be small in both the number of instances and the number of features, so the experiments are only somewhat convincing in demonstrating the method's effectiveness on larger-scale settings.

**Additional Comments:**

- Is it possible to extend the formulation to find top-k rules, rather than just a single optimal rule?
- I feel that there are many situations where it is useful to find a small number of multiple rules, but how practical is it to find just a single rule?
- (I have confirmed the revised paper, which was submitted on 21 February 2026.)

**Audience:**

Yes

**Audience Explanation:**

Yes, I think this is definitely an interesting line of work that would interest the operations research community as a potential application area, as well as the machine learning community, especially researchers working on explainable AI and HCI.
This is because it enables the design and selection of rules that are robust in practical explanation and operational processes.

**Claims And Evidence:**

Yes

**Claims Explanation:**

Majority of the claims suggested in this paper have been experimentally verified. However, point 3 under the above weaknesses raises a slight concern about the credibility of the experimental results.

**Requested Changes:**

Especially for point 1 under the above weaknesses, please consider whether improvements are possible from the following perspectives.

## For point1:
- The studies cited in this paper ([1][2]), as well as others such as [3], convert decision tree learning into ILP or MIP formulations. There appear to be many commonalities in the constraints used in these papers when formulating MIPs. If such commonalities exist, this reviewer suggests the authors mention them, and at the same time, specify the particular constraints that are either additionally introduced or modified in this paper. I belive that it will improve readability.

> [1] Dimitris Bertsimas and Jack Dunn. Optimal classification trees. Machine Learning, 106:1039–1082, 2017.

> [2]Verwer, S., & Zhang, Y. Learning Optimal Classification Trees Using a Binary Linear Program Formulation. Proceedings of the AAAI Conference on Artificial Intelligence, 33(01), 1625-1632, 2019.

> [3] Aghaei, S., Azizi, M. J., & Vayanos, P. (2019). Learning Optimal and Fair Decision Trees for Non-Discriminative Decision-Making. Proceedings of the AAAI Conference on Artificial Intelligence, 33(01), 1418-1426.

- Constraint (4) and (5) seem to represent rules by activating the rightmost leaf node when the internal node $t$ does not split. If this understanding is correct, this convention for representing it in MIP should be described before.
- Is Constraint (15) necessary? It seems that the variable constraint (27) is equivalent to this constraint.

## Minor:
- (p3, second paragraph) .. to extract rules from data. Frist, Rules Extraction => to extract rules from data. First, Rules Extraction

---

> ### Author Response · Authors · 2026-03-01
> **Response (Part 01) to Reviewer Comments**
>
> Dear Reviewer,
>
> Before we begin addressing the feedback, we deeply appreciate the time and effort you have dedicated to this manuscript, including your thoughtful comments, constructive feedback, and identification of typos we missed.
>
> ***Reviewer Comment 01*** - The MIP formulation is somewhat difficult to follow and validate, because it relies on a large number of inequality constraints and variables, which makes it hard for the reader to understand and verify the correctness of the encoding.
>
> ***Reviewer Comment 02*** - (Point1 on Requested Changes) The studies cited in this paper ([1][2]), as well as others such as [3 ... If such commonalities exist, this reviewer suggests the authors mention them, and at the same time, specify the particular constraints that are either additionally introduced or modified in this paper. I believe that it will improve readability.
>
> **Response** The large number of inequality constraints and variables naturally comes from the optimization-based tree method itself and is therefore difficult to avoid. Instead, the newly added sentence below, based on your feedback, explains which constraints are inherited from prior work and which are newly introduced. We hope this improves the readability of the MIP formulation.  On page 9 (right after the MIP formulation):
>
> *Our OPDT formulation adopts three fundamental constraints from \citet{bertsimas2017optimal}'s OCT: tree structure constraints (split decisions at branching nodes), sample routing constraints (directing samples from root to leaf), and misclassification constraints (counting misclassified samples at each leaf node). On top of these, we introduce constraints that force samples to rightmost leaf nodes when parent nodes do not split, correcting sample routing behavior for accurate VI evaluation at branching-terminated parent nodes, and constraints that linearize the logical condition for maximal VI selection among leaf nodes.*
>
> ***Reviewer Comment*** - The paper includes only limited theoretical discussion of the MIP formulation, such as how the number of decision variables and constraints scales with the number of instances or features, which makes it difficult to assess the scalability of this MIP problem from a theoretical perspective.
>
> **Response** We have added the following sentence after the MIP formulation on page 9.
>
> *For a fixed tree depth $D$, the formulation scales linearly in both $|N|$ and $|P|$, but exponentially in $D$, an inherent characteristic shared with OCT \citep{bertsimas2017optimal}. Since the new constraints and variables introduced in OPDT add only $O(2^D)$ additional variables and constraints, which are dominated by the OCT routing constraints, they do not alter the asymptotic complexity. Therefore, OPDT maintains the same $O(2^D \cdot (|P| + |N|))$ model size complexity as OCT.*
>
> ***Reviewer Comment*** - Constraint (4) and (5) seem to represent rules by activating the rightmost leaf node when the internal node does not split. If this understanding is correct, this convention for representing it in MIP should be described before.
>
> **Response** We recognize that Constraints (4) and (5) may not be immediately intuitive to the reader without prior context. To address this, we have added the following sentence before the MIP formulation, which also serves as the response to the first feedback point:
>
> *we introduce constraints that force samples to rightmost leaf nodes when parent nodes do not split, correcting sample routing behavior for accurate VI evaluation at branching-terminated parent nodes,*
>
> ***Reviewer Comment*** - Is Constraint (15) necessary? It seems that the variable constraint (27) is equivalent to this constraint.
>
> **Response** Thank you for identifying this duplication. While revising the manuscript in response to another reviewer's comments, we inadvertently introduced a duplication when adding the new type-defining constraints (18)--(28). We have removed Constraint (15) in the revised manuscript.
>
> ***Reviewer Comment*** - (p3, second paragraph) .. to extract rules from data. Frist, Rules Extraction => to extract rules from data. First, Rules Extraction
>
> **Response** Thank you for finding the typo we didn't recognize. We fix "Frist" -> First".

---

> ### Author Response · Authors · 2026-03-01
> **Response (Part 02) to Reviewer Comments**
>
> ***Reviewer Comment*** - Most of the benchmarks appear to be small in both the number of instances and the number of features, so the experiments are only somewhat convincing in demonstrating the method's effectiveness on larger-scale settings.
>
> **Response** First, we want to point out that the runtime of OPDT includes the time required to prove optimality, unlike other heuristic methods, which only find a solution without any optimality guarantee. Generally, the scalability limitation is common to all optimization-based methods. However, we believe that we demonstrate the runtime could be reasonable if prior knowledge is effectively utilized throughout this paper. For example, in Table 4, we use only data type (numerical or categorical) knowledge to split the problem. The Chronic Kidney dataset demonstrates that runtime can decrease substantially from approximately 1,000 seconds to within 20 seconds by exploiting this simple feature grouping, although this speedup is not universal across all datasets. Practically, we believe that similar speedups are achievable for real-world datasets through the BSC framework using prior domain knowledge, even though there remains a possibility of failure in the worst-case scenario.
>
> ***Reviewer Comment*** - I feel that there are many situations where it is useful to find a small number of multiple rules, but how practical is it to find just a single rule?
>
> **Response** Let us explain a virtual business case in which OPDT is useful, since we cannot share a real case for security reasons. Suppose that an attacker compromises our system, and the system has an alarm mechanism to detect abnormal spikes through an aggregate-level monitoring tool. In such a scenario, it is possible to identify that certain data around a specific time originates from an attacker's activity. If the monitoring system is sufficiently responsive, the volume of attacker activity is unlikely to reach millions of records before detection. Roughly, the volume is on the order of hundreds to thousands of instances (free from the scalability limitations of OPDT). Moreover, an attacker's behavior is generally, though not necessarily, characterized by a single pattern that exploits a specific system vulnerability (which justifies a single-rule extraction algorithm).
>
> When an attack occurs, we must immediately defend our system by understanding the attacker's behavior across the available features in our system. This action should be interpretable with certainty by human decision makers (which motivates the use of interpretable AI).
>
> Also, the action should be highly reliable. For example, suppose that an attacker's activity originates from a specific area. A straightforward way to prevent it would be to block all transactions from that area. However, this action creates follow-up problems for other good users who reside in the same area. It is therefore necessary to tailor the action to filter out good users from the prevention scope. This motivates us to design the BSC framework for compliance (e.g., if there are regulations we must follow) or prior knowledge integration (e.g., [ZIP codes around the area, NY10001, NY10002, etc.] AND [what other conditions could help distinguish the attacker from good users?], so we want to identify conditions in addition to the area condition). Obviously, any action must minimize false positives, as incorrect actions on non-attackers incur substantial financial loss (which motivates the optimization-based approach).
>
> These (thousands-level data, single-pattern attacks at a time, interpretability, prior knowledge, and optimality) are the core motivations of this paper, which has not been previously researched prior to this work.
>
> ***Reviewer Comment*** - Is it possible to extend the formulation to find top-k rules, rather than just a single optimal rule?
>
> **Response** Given the business case described above, we first focus on extracting a single optimal pattern from data. As ongoing future work, we are developing a new MIP formulation to identify a $k$-rule set that maximizes coverage while minimizing misclassification; however, this approach differs fundamentally from the formulation presented in this paper. We therefore believe it should be presented as a separate paper rather than as part of the current submission. We believe the current paper, which identifies a single optimal rule, is valuable.
>
> However, if we relax optimality, as a simple heuristic extension, one could call OPDT $k$ times sequentially, removing data matching the previously identified pattern at each iteration. We do not include this approach, as it is a heuristic idea that would undermine the paper's claim on the optimality and interpretability of OPDT. However, for the attacker case described above, if the attacker employs multiple patterns, this naive iterative approach could detect multiple patterns in the data sequentially. Therefore, OPDT remains useful for the attack with multiple patterns.

---

> > ### Comment · Reviewer_ZhSs · 2026-03-10
> >
> > Thank you for revising the manuscript. I also appreciate the clarifications, especially regarding the motivation for identifying a single optimal rule. I am now convinced that there are important real world applications for this setting, and all of my concerns have been resolved.

---

### Review · Reviewer_by7T · 2026-02-26

**Summary Of Contributions:**

**summary**

This paper proposes an optimal classification pattern discovery method using structural constraints based on prior knowledge.
The method is built on the concept of optimal decision trees and is solved using Mixed-Integer Programming (MIP).
The prior knowledge, which consists of grouping related features and applying length constraints, is designed to be intuitive for humans and helps reduce the search space.
In the experiments, the proposed method is compared with conventional pattern discovery techniques using various datasets, and its effectiveness is discussed.

**strengths**
* Previous literature on rule-based machine learning and optimal decision trees is well-summarized.
* A new pattern evaluation metric, which was not present in conventional research, is proposed.
* The approach is intuitive and clear.

**weakness**
* The experimental setup and discussion are insufficient for the claims made.
* The proposed method lacks scalability.
* The proposed method can only discover a single pattern.

**Additional Comments:**

* Since there is no mention of structural constraints (prior knowledge) in the title or the abstract, it is confusing while reading the main text.
* Regarding the end of the second paragraph in Section 2.1, a technical reference is cited for a practical topic. Please verify if this is the correct citation.
* Is the third sentence of the third paragraph in Section 2.1 a typo (prefect -> perfect)?

**Audience:**

Yes

**Audience Explanation:**

Since pattern discovery is a fundamental element of machine learning and the paper also addresses the recent trend of optimal decision trees, I believe there are a sufficient number of interested readers.

**Broader Impact Concerns:**

This paper has no ethical concerns.

**Claims And Evidence:**

No

**Claims Explanation:**

While the basic goals and ideas of this paper are clearly supported, the discussion and experiments are insufficient for some of the specific claims. Details are provided in the recommendations under **Requested Changes**.

**Requested Changes:**

**recommendation**
* While the authors claim that feature grouping can be automated using ML, the experiments only employ simple numerical-categorical grouping. Although this may be due to scalability issues, if the authors state that automation is possible, they should provide at least one specific example of an automated feature grouping process and conduct experiments using it.
* The parameter $w$ used in the experiments does not appear to be explicitly stated. Furthermore, a sensitivity analysis should be performed to show how the coverage and precision of the rules obtained by the proposed method change with variations in $w$.
* Table 8 shows not only the superior aspects of OPDT but also that it can easily overfit on training dataset; however, there is no mention of this.

**strengthen**
* The persuasiveness of the proposed method would be enhanced by providing visual examples showing how the patterns obtained by the proposed method differ from those of competing methods.
* Since the proposed method finds only a single pattern, it is often insufficient as a classifier. In general, it is desirable to extract multiple different patterns. This could be easily achieved by repeatedly applying the proposed method to uncovered samples; do the authors envision such a use case? If so, the addition of relevant experiments is desired.

---

> ### Author Response · Authors · 2026-03-02
> **Response (part 01) to Reviewer Comments**
>
> We appreciate the comments and recommendations to enhance our manuscript. We address each request as follows:
>
> ***Reviewer Comment*** - The persuasiveness of the proposed method would be enhanced by providing visual examples showing how the patterns obtained by the proposed method differ from those of competing methods.
>
> **Response** Thank you for a feedback about a visual example. We've add a visual exmaple to show how to OPDT imrpove VI on page 12:
>
> ***Reviewer Comment*** - Since there is no mention of structural constraints (prior knowledge) in the title or the abstract, it is confusing while reading the main text.
>
> **Response** We have added the following setence in the abstract.
>
> *To incorporate prior knowledge and compliance requirements, we further introduce the branching structure constraints (BSC) framework, which enables decision makers to embed domain knowledge and constraints directly into the model.*
>
> ***Reviewer Comment*** - Regarding the end of the second paragraph in Section 2.1, a technical reference is cited for a practical topic. Please verify if this is the correct citation.
>
>
> **Response** Thank you for identifying the wrong citation. We intended to add a domain-specific citation for each application. We missed updating the LaTeX citation for each case properly. We have revised the sentence accordingly as follows:
>
> *Rule-based classifiers have already demonstrated successful application across diverse domains, including medical diagnosis (Asgari et al, 2019), financial fraud detection (Ali et al, 2022), intrusion detection (Lee et al, 1999), and machine failure diagnosis in manufacturing systems (Jiang'hong & Xial'li, 2009).*
>
> ***Reviewer Comment*** - Is the third sentence of the third paragraph in Section 2.1 a typo (prefect -> perfect)?
>
> **Response** Thank you for identifying the typo. We have corrected "prefect" to "perfect" in the revised manuscript.
>
>
> *PRISM (Cendrowska 1988) introduced a unique approach of generating rules by selecting examples of a specific class and iteratively adding conditions until the obtained rule has perfect precision.*

---

> ### Author Response · Authors · 2026-03-03
> **Response (part 02) to Reviewer Comments**
>
> We appreciate the reviewer's time and careful reading of our manuscript. We would like to respectfully clarify that several of the recommendations appear to extend beyond the claimed scope of this paper. We address each point in turn below.
>
> ***Reviewer Comment*** - The experimental setup and discussion are insufficient for the claims made.
>
> **Response** We appreciate this feedback. We hope that our responses to the specific comments in the Requested Changes section address the concerns. If this concern refers to a specific claim or section not addressed by the above responses, we would greatly appreciate more specific guidance to improve the manuscript.
>
>
> ***Reviewer Comment*** - The proposed method lacks scalability.
>
> **Response** The scalability issue is inherent in all optimization-based tree models, as the underlying problem is NP-hard. Note that the runtime of this method includes the time required to prove optimality, unlike prior heuristic methods. We believe there are high-stakes domains where a high-quality solution is required rather than a quickly found one, and we claim this is the purpose of this paper.
>
> The claim of this paper is not to find an algorithm to resolve NP-hard problems theoretically. Rather, we aim to provide empirical evidence that an optimization-based method can resolve certain pattern detection problems when prior knowledge is integrated into the MIP formulation. This paper provides convincing computational results showing that the runtime is at a reasonable level on moderately sized datasets, unlike OCT, which requires hours for classification. To overcome and improve the runtime issue of optimization-based methods, we provide several new ideas and tips in the "Performance Enhancements" section and demonstrate the computational gain in Table 5 with various configuration tests. Furthermore, we have confirmed that the time to solution (i.e., the point at which the solver first improves upon a heuristic solution, without necessarily reaching optimality) is much faster than the total runtime, as shown in Table 7, demonstrating its practical value in improving the quality of a prior solution toward the optimal one.
>
> Also, please, check my response to other reviewers:
>
> - Regarding Computational Overhead section under "Response (part 02) to Reviewer Comments" titled response
>
> - First, we want to point out that the runtime of OPDT includes the time required to prove optimality, unlike other heuristic methods, which only find a solution without any optimality guarantee. ...
>
>
> ***Reviewer Comment 01*** - The proposed method can only discover a single pattern.
>
> ***Reviewer Comment 02*** - Since the proposed method finds only a single pattern, it is often insufficient as a classifier. In general, it is desirable to extract multiple different patterns. This could be easily achieved by repeatedly applying the proposed method to uncovered samples; do the authors envision such a use case? If so, the addition of relevant experiments is desired.
>
>
> **Response** We would like to clarify that the goal of this paper is not to build a general-purpose classifier, but rather to find a single optimal pattern in data — a fundamentally different and purposefully scoped problem. The title of this paper uses "pattern," not "ruleset" to reflect this distinction explicitly.
>
> The motivation for focusing on a single pattern arises from a concrete business case, as described in our response to another reviewer regarding the "virtual business case", where positive samples originate from a single reason or action rather than a mixture of distinct activities. In such cases, any second or third pattern found in the data (after mixing random negative samples into the target positive class) would likely represent noise introduced by the randomness of the negative samples. Existing RBML methods fail to identify a single high-quality pattern for such positive samples, as shown in Tables 6 and 8 compared to OPDT.
>
> We therefore designed OPDT to find a single optimal pattern on purpose, as there was no prior method that could do so with optimality guarantees. We believe that finding a single optimal pattern in data is no less valuable than finding multiple patterns — they are simply different problems with different objectives. To the best of our knowledge, there is no prior work on finding an optimal pattern in data.
>
> Please, check my response about other reviewer's following feedback:
>
> - I feel that there are many situations where it is useful to find a small number of multiple rules, but how practical is it to find just a single rule?
>
> - Is it possible to extend the formulation to find top-k rules, rather than just a single optimal rule?

---

> > ### Comment · Reviewer_by7T · 2026-03-03
> > **Follow-Up Comments**
> >
> > Thank you very much for your careful response.
> >
> > **Regarding the comment, “The proposed method lacks scalability.”**
> >
> > I understood this point as an unavoidable weakness inherent to the method, and it was not intended as a criticism of the main claim of the paper. I apologize if my wording caused any misunderstanding. With the detailed explanation provided, I was able to better appreciate the motivation for guaranteeing optimality.
> >
> > **Regarding the Comment 01 and Comment 02**
> >
> > Your additional explanation helped me to clearly understand the importance of discovering a single optimal pattern. However, in the manuscript, the explanation places emphasis on the relationship between the proposed approach and RBML, and the discussion on the usefulness of a single pattern is insufficient. As a result, the manuscript makes it easy to become confused as to whether the paper aims to propose a generally useful method for RBML to discovery "patterns" or whether it strongly focuses on the discovery of a valuable single "pattern". It is recommended to make use of the example explanation that was presented to the other reviewer to further polish and improve the manuscript.

---

> > > ### Author Response · Authors · 2026-03-04
> > > **Response to Reviewer Follow-up (Part 01) Comments**
> > >
> > > About the second follow-up questions, they require certain major revisions to the experiments. We are still considering how to address or reply to them. It will take a few days.
> > >
> > > ***Reviewer Comment*** Regarding the comment, “The proposed method lacks scalability.”
> > >
> > > **Response** Thank you for your understanding of the nature of optimization-based models. The scalability concern has always been the primary reason for rejecting this type of paper from many ML conferences. For this reason, we had originally planned to submit to traditional operations research journals to avoid rejection. However, as a final attempt, we submitted to the ML community journal process to obtain rigorous feedback and to determine whether this method is acceptable to the broader ML community. We appreciate your careful reading of our paper and your thoughtful responses.
> > >
> > >
> > > ***Reviewer Comment*** Regarding the Comment 01 and Comment 02
> > >
> > > **Response** Thank you for your valuable feedback from a reader's perspective. We genuinely appreciate your insight into why a reader may find the motivation unclear. Due to internal security policy, we are not permitted to use any internal data or specific cases that exactly match this method in the manuscript. Instead, we have used broad domain examples — such as epidemic outbreaks, credit fraud, and intrusion detection — to illustrate the motivation. We hope you understand this constraint.
> > >
> > > * First, we have added "single" to the abstract explicitly and added new sentences to clarify the paper's scope. Also, we have changed "extract" to "discover":
> > >
> > > [Abstract]
> > > *This paper introduces the optimal pattern detection tree (OPDT) for binary classification ... extract an optimal pattern from data -> This paper introduces the optimal pattern detection tree (OPDT), a rule-based machine learning model based on novel mixed integer programming to discover a single optimal pattern in data through binary classification.*
> > >
> > > * Second, we have added the following new sentence and modified the existing sentence (changing "for binary classification" to "through binary classification").
> > >
> > > [Introduction] (Right after introducing RBML papers and its value)
> > > *However, existing RBML methods are designed to extract a ruleset (i.e., a collection of multiple rules) rather than a single optimal pattern in data. In many real-world scenarios, such as an epidemic outbreak, credit fraud, or intrusion detection, positive samples originate from a single underlying cause, and a single high-precision rule is both sufficient and more actionable than a complex ruleset. To the best of our knowledge, no prior method has conducted research on extracting a single optimal pattern.*
> > >
> > > *"This paper proposes a novel rule extraction algorithm for binary classification that adheres to user-defined structural constraints, ensuring high interpretability while incorporating prior domain knowledge."* ==> *"This paper proposes a novel algorithm to extract a single rule through binary classification that adheres to user-defined structural constraints, ensuring high interpretability while incorporating prior domain knowledge."*

---

> ### Author Response · Authors · 2026-03-03
> **Response (part 03) to Reviewer Comments**
>
> ***Reviewer Comment*** - The parameter w used in the experiments does not appear to be explicitly stated. Furthermore, a sensitivity analysis should be performed to show how the coverage and precision of the rules obtained by the proposed method change with variations in w.
>
> **Response** Thank you for the feedback. We will consider this experiment for an extended version. The reason we did not test sensitivity on the parameter $w$ is that it is more related to the characteristics of the data (e.g., whether a dataset contains a rule with 0.9 precision or 0.99 precision, or the trade-off between volume and precision) rather than the characteristics of the OPDT method itself (e.g., optimality, impact on runtime, interpretability, flexibility of BSC, and speedup by feature grouping). We believe that the characteristics of OPDT remain consistent across different values of $w$.
>
>
> ***Reviewer Comment*** - While the authors claim that feature grouping can be automated using ML, the experiments only employ simple numerical-categorical grouping. Although this may be due to scalability issues, if the authors state that automation is possible, they should provide at least one specific example of an automated feature grouping process and conduct experiments using it.
>
>
> **Response** We appreciate this valuable feedback. We will consider designing an experimental case for the follow-up research. We acknowledge that the experiments in this paper employ only numerical-categorical feature grouping, rather than an ML-automated grouping approach. We would like to clarify our rationale for this choice.
>
> In the title, we use the term "optimal"; therefore, we wanted to exclude any heuristic idea that violates optimality in this version of the paper. A computational case that uses feature grouping based on heuristic ML output (e.g., selecting only the top-$k$ features from hundreds of features) would not support optimality, even though the computational time would be faster.
>
> We do not know exactly which sentence led you to believe we claim that feature grouping can be automated using ML, but if the following sentence is the source of that interpretation, we would like to respectfully clarify that our statement — "Even when domain knowledge is unavailable, various ML methods can extract prior information about features" — does not claim that feature grouping can be automated by ML algorithms. Translating such information into feature groups still requires human judgment. We simply emphasize that prior information about features is universally accessible even without specialized domain expertise (e.g., without being a medical doctor for healthcare data), through methods such as SHAP, Boruta, and feature importance from ML decision trees, as described in Section 3. The sentence states only that ML methods can extract prior information about features, not that this process constitutes automation. We hope this clarifies the intended scope of this paper.
>
> With a similar rationale, we do not include a heuristic approach to extracting $k$ rules by calling OPDT $k$ times sequentially, as it does not align with the optimality guarantee.
>
>
> ***Reviewer Comment*** - Table 8 shows not only the superior aspects of OPDT but also that it can easily overfit on training dataset; however, there is no mention of this.
>
> **Response** We appreciate this observation. We would like to clarify the distinction between optimization and overfitting in the context of OPDT. Overfitting in tree-based ML algorithms generally occurs when the tree grows excessively deep, creating branches that explain individual training instances (e.g., a very deep tree that perfectly classifies every training sample in a leaf node). This is typically verified by a large gap between training and test performance. However, as shown in Table 8, OPDT shows comparably strong performance on out-of-sample test data. Furthermore, even in cases where OPDT does not rank in the top 4, none of the cases produce the worst result among the other methods (if the OPDT method inherently creates an overfitted model, it would produce poor test performance).

---

> ### Comment · Reviewer_by7T · 2026-03-03
> **Follow-Up Comments**
>
> **Sensitivity Analysis**
>
> I raised this comment precisely with the intention of examining the trade-off between coverage and precision. Even if the characteristics of the dataset do not lead to particularly insightful results, I believe this experiment is necessary to understand the role and significance of the parameter $w$.
>
> **Feature Grouping**
>
> Most grouping strategies, including numerical–categorical feature grouping, have heuristic aspects. I understand that the optimality claimed in this paper is not global, but rather holds over a heuristically restricted search space defined by the given prior knowledge. I also believe that, as long as a heuristic has a certain level of justification or expectation, the optimal pattern found within that space can be sufficiently meaningful.
>
> I understood that the paper does not suggest fully automated feature grouping. However, it is stated that prior information derived from machine learning can be utilized. Experiments that demonstrate concrete ways of leveraging such prior information would be extremely beneficial for users who wish to apply the proposed method. On the other hand, numerical–categorical feature grouping lacks a priori justification. An experimental setting that better brings out the appeal of the proposed method should be adopted.
>
> **Overfitting**
>
> In the four datasets—Chronic Kidney, Echocardiogram, Heart Disease, and Thoracic Surgery—OPDT shows clearly weaker values. While these are not the worst among the compared methods, they are sufficient to raise suspicion of overfitting. Furthermore, in Tables 6 and 8, only VI is reported as the evaluation score; however, by definition, the scale of VI changes depending on the sample size, which makes comparison between training and test sets difficult. This also appears to contribute to my suspicion of overfitting. Therefore, it would be better to base the discussion on Table 9, in which coverage and precision are also reported separately.

---

> > ### Author Response · Authors · 2026-03-08
> > **Response to Reviewer Follow-up (Part 02) Comments**
> >
> > ***Reviewer Comment*** - Sensitivity Analysis
> >
> > **Response** We have added Table 9 to show the trade-off between coverage and precision by parameter $w$ with the following sentence:
> >
> > - *Finally, Table 9 presents the sensitivity analysis of the weight parameter $w$ on the UCI Hepatitis dataset. As $w$ decreases from 10 to 2, OPDT exhibits a monotone precision–coverage trade-off, yielding higher coverage at the cost of reduced precision, confirming that $w$ functions as an adjustable precision requirement. Runtime remains stable across all $w$ values, and test performance closely tracks training performance, indicating strong out-of-sample performance.*
> >
> > ***Reviewer Comment*** - Feature Grouping
> >
> > **Response** Thank you for recognizing the potential of OPDT in combination with ML methods. We have added Figure 4 and Table 11 to demonstrate a way to use feature importance from LGBM as prior information. The demonstration shows the computational time to optimality is improved by feature grouping, as it is heuristic by nature. Since this is for demonstration purposes, we have placed it in the Appendix, as we feel this heuristic approach is not well aligned with the other experiments in the main body. We hope this is acceptable. However, we agree that a concrete example showing how to connect this paper with ML methods is very helpful for follow-up research. We think this combination approach requires more careful exploration as independent research.
> >
> > - *Feature Grouping Using ML-Based Feature Importance: As a concrete demonstration that prior information extracted from ML can help feature grouping, we select the German dataset, which is the largest dataset among those in Table 3. Suppose that we need to find a pattern complying with the {categorical}--{numerical} structure. First, feature importance is extracted from a LightGBM (LGBM) model (Ke et al., 2017) trained with 1,000 boosting iterations. Figure 4 presents the feature importance scores evaluated by two complementary metrics: information gain and split count. Based on this prior information, we select the top six features from each metric, as highlighted in red in Figure 4, to define two feature groups. Table 11 shows the VI improvement over time with a runtime limit of 600 seconds. As shown in Table 11, OPDT with LGBM reaches the optimal VI of $41.0$ in 5 seconds, whereas OPDT Only reaches the same optimal VI in 23 seconds, and reduces the total runtime to prove optimality from $446.4$ seconds to $90.8$ seconds, which is a fivefold reduction in runtime. This experiment demonstrates a heuristic approach in which an ML method is used to extract feature importance rankings to define feature groups for the BSC framework.*
> >
> > ***Reviewer Comment*** - Overfitting
> >
> > **Response** We have enhanced the paragraph to explain Table 9 in detail. In particular, for the weak cases, we analyzed the likely reasons why they occurred. This discussion naturally connects to the sensitivity analysis of parameter $w$. In future work, we believe that more extended experiments are needed, including a proper $w$ setup for each dataset.
> >
> > - *However, it is worth noting that OPDT's strong training performance does not always transfer to the test set. For example, even though OPDT achieves a higher VI value than BSCCART on training data for the German and Heart Disease datasets, it fails to show better VI on test data. Specifically, in the German dataset, OPDT achieves precision $0.936$ and coverage $0.225$ on training data, but precision $0.883$ and coverage $0.231$ on test data, resulting in a test VI of $-7.9$. In contrast, BSCCART achieves precision $0.920$ and coverage $0.315$ on training data, and precision $0.896$ and coverage $0.326$ on test data, resulting in a test VI of $-1.8$. A similar pattern is observed in the Heart Disease dataset: OPDT achieves precision $0.979$ and coverage $0.165$ on training data, but precision $0.800$ and coverage $0.180$ on test data, yielding a test VI of $-9.2$, [...]. In both cases, OPDT identifies a narrower region (lower coverage than BSCCART on training data) to achieve higher precision. However, this precision advantage is not stable on the test set. This suggests that the weight parameter $w$ may force OPDT toward a narrow region that fails to represent the whole dataset, implying that $w = 10$ may not be an appropriate value for representing the dominant pattern in these datasets. As another case, in the Thoracic Surgery dataset, [...] the weight parameter $w$ penalizes OPDT's broader coverage rule. This further supports the observation that the choice of $w$ is critical to finding a representative pattern in data. Additionally, it is worth noting that IDS and PRISM are not constrained to rules with only two conditions, unlike OPDT, potentially giving them an advantage in searching for patterns having more than two conditions.*

---

> > > ### Comment · Reviewer_by7T · 2026-03-09
> > >
> > > Thank you for carefully revising the manuscript!
> > > Although it is a very minor point, please add a sentence specifying the default setting for $w$ used in the experiment in Section 4.2 (presumably $w=10$?).
> > > I believe the future challenges have also become clearer, so if you could appropriately organize them in the Conclusion and Future Work section, all my concerns would be resolved.

---

> > > > ### Author Response · Authors · 2026-03-09
> > > > **Response to Reviewer Follow-up (minor) Comments**
> > > >
> > > > ***Reviewer Comment*** - add a sentence specifying the default setting for $w$.
> > > >
> > > > **Response** Thank you for catching this. We thought we mentioned it in the manuscript under Section 4.3 (Computational Experiments), but we recognize that we failed to state its value explicitly next to $w$ in that section. In addition, we have added a new sentence at the beginning of Section 4 to make it clear that we use 10 for all experiments (except for the sensitivity analysis).
> > > >
> > > > - On page 14: *VI is computed based on volume and misclassified samples with respect to weight $w$.* ==> *VI is computed based on volume and misclassified samples with respect to weight $w = 10$.*
> > > >
> > > > - Section 4 (Computational Results): *In all experiments, we set the weight $w$ for the VI index to 10 as the default value.*
> > > >
> > > > ***Reviewer Comment*** - Conclusion and Future Work section
> > > >
> > > > **Response** Thank you for your valuable feedback. By the help of the reviewer's comments, we are able to identify clear future research directions. We deeply appreciate the insightful feedback that has helped us recognize the direction of future research to address these open challenges. We have added the following sentences for future work.
> > > >
> > > > - *Despite the contributions presented in this paper, a number of important challenges remain to be addressed in future work. First, even though OPDT can find an optimal pattern given a specific rule structure and weight, it requires a decision maker to specify these parameters appropriately to identify a representative pattern in data that generalizes beyond the training data. Second, while OPDT is designed to extract a single optimal pattern, it is natural to extend it to discover a set of rules that represent data optimally, which remains an important direction for future work. Finally, by the inherent nature of optimization-based methods, OPDT still has a scalability limitation on large-scale data. As demonstrated by the speedup achieved through BSC with different feature groupings, further research is needed to develop systematic approaches for utilizing the BSC framework with prior knowledge tailored to specific datasets.*

---

> > > > > ### Comment · Reviewer_by7T · 2026-03-10
> > > > >
> > > > > Thank you for revising the manuscript!

---

### Decision · Action_Editor_zhEp · 2026-03-30

**Recommendation:** Accept as is

**Additional Comments:**

The reviewers raised several critical points regarding the initial submission:
* Experimental Validation: The need for a more thorough investigation into automated feature grouping, sensitivity analysis of the weight parameter, and a dedicated discussion on overfitting.
* MIP Formulation: Insufficient clarification regarding the novelty of the Mixed-Integer Programming (MIP) constraints compared to existing ILP/MIP-based decision tree methods, and technical ambiguities in certain constraints.
* Evaluation Metrics: The lack of a clear comparison between the proposed Volume-Impurity (VI) index and established rule-evaluation metrics.
* Computational Efficiency: Concerns regarding scalability and the potential for improving efficiency by decomposing the problem.

The authors have addressed most of the reviewers' concerns in the revised manuscript.
While the novelty of the MIP formulation itself is incremental and scalability remains a challenge, the core strength of OPDT lies in its ability to incorporate human prior knowledge through structural constraints (e.g., feature grouping).
The extensive experiments demonstrate that the method provides clear utility and competitive performance on several benchmarks.
Given the novelty of the problem setting and the rigorous evaluation, this work meets the criteria for TMLR.

**Audience:**

Yes

**Audience Explanation:**

Optimal decision trees and pattern detection are important topics in machine learning, and is of interest to TMLR readers.

**Claims And Evidence:**

Yes

**Claims Explanation:**

This paper introduces Optimal Pattern Detection Trees (OPDT), a rule-based binary classification framework that leverages Mixed-Integer Programming (MIP) to discover optimal symbolic patterns.
The proposed method can incorporate structural constraints based on prior knowledge, such as grouping related features.
OPDT maximizes the Volume-Impurity (VI) index, newly introduced in this paper as the number of training examples covered minus a weighted penalty for misclassifications.
In OPDT, Users can define feature groups and constraints on which features can be queried at specific tree nodes, making the resulting rules more intuitive and compliant with domain requirements.
Experiments across various datasets demonstrate that OPDT achieves comparable or superior performance to conventional pattern discovery baselines while maintaining optimality guarantees and reasonable runtimes.